# CaptainCook4D: A Dataset for Understanding Errors in Procedural Activities

**Rohith Peddi** + *   **Shivvrat Arya** +   **Bharath Challa** +   **Likhitha Pallapothula** +

**Akshay Vyas** +   **Bhavya Gouripeddi** +   **Qifan Zhang** +   **Jikai Wang** +

**Vasundhara Komaragiri** +   **Eric Ragan** ×   **Nicholas Ruozzi** +   **Yu Xiang** +   **Vibhav Gogate** +

## Abstract

Following step-by-step procedures is an essential component of various activities carried out by individuals in their daily lives. These procedures serve as a guiding framework that helps to achieve goals efficiently, whether it is assembling furniture or preparing a recipe. However, the complexity and duration of procedural activities inherently increase the likelihood of making errors. Understanding such procedural activities from a sequence of frames is a challenging task that demands an accurate interpretation of visual information and the ability to reason about the structure of the activity. To this end, we collect a new egocentric 4D dataset **CaptainCook4D** comprising 384 recordings (94.5 hours) of people performing recipes in real kitchen environments. This dataset consists of two distinct types of activities: one in which participants adhere to the provided recipe instructions and another in which they deviate and induce errors. We provide 5.3K step annotations and 10K fine-grained action annotations and benchmark the dataset for the following tasks: error recognition, multi-step localization and procedure learning[2].

## 1 Introduction

*Have you ever excitedly prepared your favourite meal after a long day, only to be disappointed upon realizing you missed a key ingredient?* Such scenarios are common because performing long step-by-step procedures increases the likelihood of making errors. While some errors are harmless and can be corrected with little consequence, others can have detrimental consequences, particularly those that occur during medical procedures or complex chemical experiments. Therefore, there is a pressing need to build AI systems that can guide users in performing procedural activities [15].

A key problem we need to solve in order to build such AI systems is **procedural activity understanding**, a challenging and multifaceted task that demands **interpreting what is happening** —specifically, determining whether the person is following the procedure correctly or making an error, **anticipating what will happen**, and **planning the course of action** to accomplish the goal. For effective interpretation, the system must be capable of recognizing and categorizing actions while assessing the current state of the environment. To anticipate what might happen next, it should be able to forecast actions right from the start of the interaction or even before it begins. Additionally, planning a course of action necessitates understanding the potential consequences of these actions. Numerous datasets have been developed to improve our understanding of procedural activities. However, these datasets only include videos of individuals performing step-by-step tasks correctly without making any errors.

---

* Corresponding Author , + = UT Dallas, × = University of Florida

[2]website: https://captaincook4d.github.io/captain-cook/

38th Conference on Neural Information Processing Systems (NeurIPS 2024) Track on Datasets and Benchmarks.

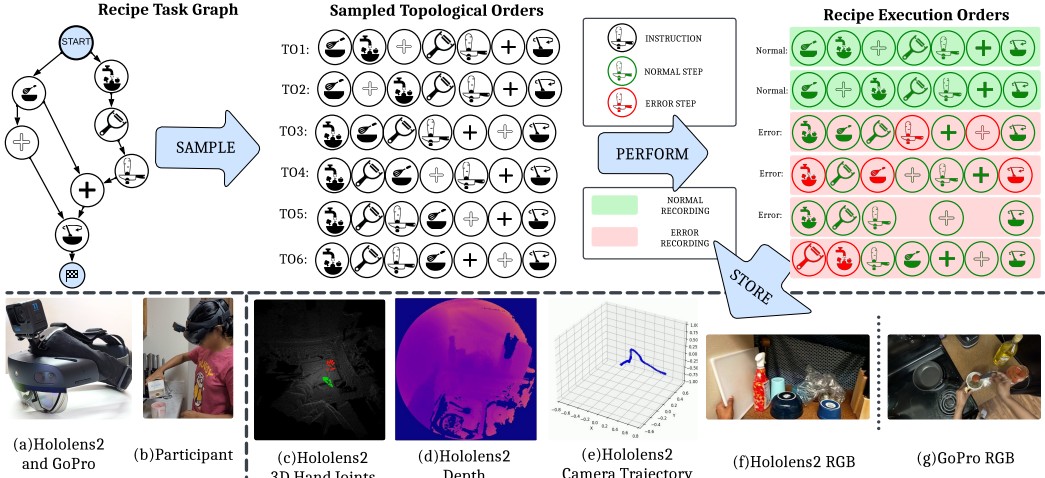

Figure 1: **Overview.** Top: We constructed task graphs for the selected recipes. These graphs facilitated sampling topological orders (cooking steps) that participants followed to perform. During the execution of these steps, participants induced errors that were both **intentional** and **unintentional** in nature. Bottom Left: We present the sensors employed for data collection. Bottom Right: We describe the details of the modalities of the data collected while the participant performs the recipe.

But, for AI systems to effectively identify errors in procedural activities, it is essential to have datasets that include both normal and error videos along with corresponding error annotations (descriptions).

**Contributions.** We introduce an egocentric[3] 4D dataset designed to enhance AI systems' understanding of procedural activities and improve their ability to recognize and anticipate errors.

– Our dataset features participants performing recipes in real kitchen environments (Fig. 1). It includes two distinct types of activities: one where the participants follow the given recipe guidelines and another where they deviate (intentionally or unintentionally), making errors.

– We provide annotations for (a) Start and end times for each step of the recipe, (b) Start and end times for each fine-grained action/interaction for 20% of the collected data, and (c) Detailed descriptions of the errors made by participants, which allowed us to compile a comprehensive overview of different error categories along with their brief explanations.

– We provide baselines for the following procedure understanding tasks: (a) Error Recognition (supervised and zero-shot), (b) Multi-Step Localization, and (c) Procedure Learning.

## 2 Related Work

Understanding procedural activities with errors has witnessed significant traction recently and spurred the development of new datasets (see Table 1) that aid in developing novel approaches to recognize errors. Our dataset sets itself apart from others[4] by four distinctive features: (1) **Domain:** While others address errors during assembly and disassembly, we focus on cooking activities[5]. (2) **Environment:** Unlike lab environments, we collected our dataset in real kitchen environments. (3) **Multimodal capabilities**, and (4) **Error diversity**. A complete survey of all the relevant tasks is outside the scope of the paper; thus, we provide a brief review of procedure understanding tasks that are of particular interest to the proposed dataset and discuss their representative works.

**Temporal Action Localization (TAL)** in videos aims to identify and classify temporal boundaries of action instances in long video sequences. TAL methods can be categorized into two primary approaches: two-stage and single-stage. Two-stage methods operate in a sequential manner by initially generating action proposals and subsequently classifying them. In contrast, single-stage methods streamline the process by simultaneously performing action localization and classification,

---

[3]An egocentric view despite ego motions helps minimize occlusions more effectively than exo-centric videos.

[4]To the best of our knowledge, we are the first to categorize and provide brief descriptions for the error types.

[5]Cooking activities are inherently complex and encompass several types of diverse cascading and non-cascading errors that can compound and often alter the state of the environment with no point of return.

Table 1: **Ours vs Current Procedural Datasets (with/without errors)**: Our dataset not only advances the study of procedural tasks found in existing literature but also enables a systematic analysis of errors in procedures.

| Procedural | Errors | Dataset Name | Domain / Environment | Ego | Depth | Recorded | Error Labels | Errors Nature | Hours | Year |
|---|---|---|---|---|---|---|---|---|---|---|
| ✗ | ✗ | Epic-Kitchens [11] | Cooking / Real | ✓ | ✗ | ✓ | - | - | 100 | 2018 |
|  |  | Ego4D [100] | Daily-Life / Real | ✓ | ✓ | ✓ | - | - | 3000 | 2023 |
| ✓ | ✗ | 50 Salads [87] | Cooking / Real | ✗ | ✓ | ✓ | - | - | 4.5 | 2013 |
|  |  | Breakfast [51] | Cooking / Real | ✗ | ✗ | ✓ | - | - | 77 | 2014 |
|  |  | MPII Cooking 2 [78] | Cooking / Real | ✗ | ✗ | ✓ | - | - | 27 | 2015 |
|  |  | YouCook2 [103] | Cooking / Real | ✗ | ✗ | ✗ | - | - | 176 | 2017 |
|  |  | EGTEA Gaze+ [54] | Cooking / Real | ✓ | ✗ | ✓ | - | - | 29 | 2020 |
|  |  | EgoProceL [3] | Assembly / Real, Lab | ✓ | ✗ | ✓ | - | - | 62 | 2022 |
| ✓ | ✓ | EgoTV [37] | Cooking / Simulated | ✓ | ✗ | - | ✓ | Intentional | 168 | 2023 |
| ✓ | ✓ | Assembly101 [19] | Toy Assembly / Lab | ✓ | ✗ | ✓ | Partial* | Unintentional | 53 | 2022 |
|  |  | CSV [73] | Chemistry / Lab | ✗ | ✗ | ✓ | ✗ | Intentional | 11.1 | 2022 |
|  |  | HoloAssist [93] | Assembly*/ Lab | ✓ | ✓ | ✓ | ✓ | Unintentional | 166 | 2023 |
|  |  | IndustReal [80] | Toy Assembly / Lab | ✓ | ✓ | ✓ | ✓ | Int. and Unint. | 5.8 | 2024 |
|  |  | ATA [29] | Toy Assembly / Lab | ✓ | ✗ | ✓ | ✓ | Intentional | 24.8 | 2024 |
| ✓ | ✓ | **CaptainCook4D** (Ours) | Cooking / Real | ✓ | ✓ | ✓ | ✓ | Int. and Unint. | 94.5 | 2024 |

thus integrating both tasks into a single step. Datasets such as ActivityNet [18], Ava [33], Thumos14 [46], Epic-Kitchens [12], and Ego4D [32], helped develop advanced methods for TAL. Supervised Multi-Step Localization (MSL) task, while similar to TAL, specifically targets procedural datasets.

**Remark.** Our dataset, featuring both normal and erroneous actions, offers a unique perspective and helps evaluate the robustness of the MSL(TAL) methods in handling actions with deviations (errors).

**Error Recognition** in videos aims to identify errors (deviations from procedure text) in procedural activities. It was introduced as mistake detection by Assembly-101 [19] where a multi-class classification problem was formulated to classify the given clip corresponding to a procedure as correct, mistake or correction. Anomaly detection, while closely related to error recognition, differentiates itself by using static cameras and backgrounds to identify abnormal behaviour. Recently, [26] proposed an online error recognition method. Using Vision-and-Language Models (VLMs), they predict future actions and compare these predictions with actual observations to recognize errors online[6].

**Remark.** Unlike assembly, cooking involves continuous changes in the shape and color of the ingredients thus making our dataset valuable for developing error recognition methods transferable to procedural activities in the medical sector or that involve performing complex chemical experiments.

**Procedure Learning** in videos aims to identify the key steps in long video sequences and determine their logical order to complete a procedural activity. Datasets such as CrossTask [105], COIN [89], EgoProceL [3], Egtea [22], Meccano [75], Epic-Tents [44], helped develop advanced methods for supervised, weakly-supervised and self-supervised procedure learning task variants.

**Remark.** Videos in our dataset are characterized by a longer average step length, which presents a challenge for algorithms previously designed for the existing egocentric procedure learning datasets.

Other related tasks include Video Summarization [62], Temporal Action Segmentation [20, 53, 1, 8, 28], Object State Change Detection [86], Action Localization [106, 84], Adverb Recognition [10, 10], Task Verification (Sequence Verification [97]) [37], Long Video Understanding [43, 41, 100], Key-Step Localization [65, 58, 39], Procedure Planning [40] (Goal-Step Inference [99, 52, 70]), Self-supervised procedural knowledge extraction [63] (Visual Transformation Telling [95]), Sequence-to-sequence alignment [64, 35, 4, 6, 90, 47, 48, 79, 45, 88, 98, 85, 94] (Audio, Video synchronization [61, 7, 72, 83]), Scene Graph Anticipation [69, 68], Temporal Adaptation, Semantic Role Labelling [2, 5, 42], Procedure Learning [104, 49, 57, 102], Action Anticipation in Procedural Videos [81, 71].

## 3   Data Collection

**Sensors.** We utilized a Hololens2 device and a GoPro Hero 11 camera mounted on the user's head (see Fig. 1) to capture the activity data. We built a custom tool using hl2ss [14] to capture data from the depth sensor, front RGB camera, microphone and the depth sensor of the hololens2 device. We additionally captured the processed head and hand tracking information provided by the HoloLens2.

---

[6]At the time of writing, the code for [26] was not released.

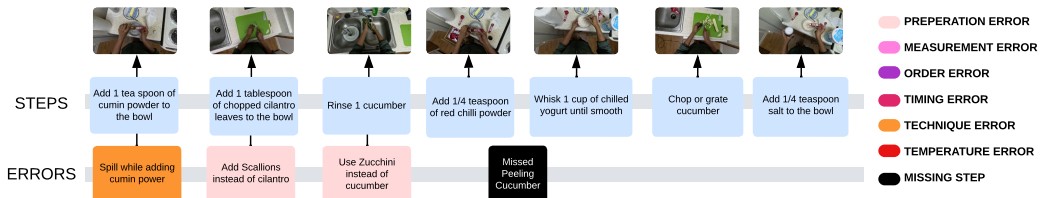

Figure 2: **Snapshots** of steps and recorded errors while preparing the recipe *Cucumber Raita*. Three of the four errors were intentional, but the participant missed the *Peeling* step **unintentionally**.

**Recipes.** We curated a selection of 24 cooking recipes sourced from WikiHow (refer to Appendix C), specifically focusing on recipes with a preparation time of 30 minutes or less. These recipes encompassed a wide range of culinary traditions, showcasing the diversity of cooking styles in various cuisines. Our primary objective was to identify and capture potential errors that could arise from using various authentic cooking instruments in preparing recipes sampled from different cuisines.

**Task Graphs.** visually represents the sequential steps required to complete a recipe. Each node in the task graph (for a recipe) corresponds to a step in a recipe, and a directed edge between a node $x$ and a node $y$ in the graph indicates that $x$ must be performed before $y$. Thus, a task graph is a directed acyclic graph, with a topological order representing a valid recipe completion. To construct task graphs for selected recipes, we identified all the critical steps involved and determined their inter-dependencies, thus establishing a topological order of tasks (see website for final task graphs).

## 3.1 Protocol

Our dataset was compiled by eight participants[7] in 10 different kitchens. Each participant was provided a tablet-based recording interface accessible through a web browser, a GoPro and a Hololens2. Participants were instructed to adjust their GoPro cameras to capture footage in 4K resolution at 30 fps to ensure high-quality video. The HoloLens2 device was programmed to stream RGB frames at 360p resolution and 30 fps. It also streamed depth frames in Articulated Hand Tracking mode, referred to as *depth_ahat* mode. Besides visual data, the device also streamed three streams of IMU (Inertial Measurement Unit) sensor data and spatial data, capturing both head and hand poses[8].

**Normal Recordings.** A recording is classified as a **normal recording** when it is captured as the participant accurately follows the procedure described in the recipe. Participants are presented with one of the pre-constructed topological orders of the selected recipe[9], as determined by the task graphs. The participants then follow and perform each step from the topological order sequentially.

**Error Recordings.** A recording is classified as an **error recording** when it is captured while the individual deviates from the recipe's procedure, thereby inducing errors. Following the terminology used in scientific disciplines such as neuroscience [9] and chemistry, we will refer to deviations from procedures as *errors*[10]. Following [9, 25, 27], we classified common errors performed during a cooking activity into the following categories: (1) Preparation Error, (2) Measurement Error, (3) Technique Error, (4) Timing Error, (5) Temperature Error, (6) Missing Steps, and (7) Ordering Errors.

**Error Induction.** We developed three strategies[11] for participants to choose from, each tailored to perform the recipe in a specific environment. After choosing the strategy, participants were given detailed instructions on how to perform the recipes. We list the strategies presented to the participants (1) **Impromptu**: Participants were asked to induce errors while performing the recipe. Following the completion of each recording, participants used a web-based interface to update the errors they performed during each step. Due to the complex nature of cooking activities and the lack of experience of the participants in cooking, many errors induced in this strategy were **unintentional** (Figure 2 presents one such example). (2) **Disordered Steps**: Participants were given pre-prepared error scripts with missing steps and ordering errors. (3) **Induct Error**: Participants used a web-based

---

[7]During filming, participants were instructed to be alone in the kitchen environment and remove any items that could potentially identify them, such as personal portraits, mirrors, and smartwatches with portraits.

[8]The University of Florida IRB approved our protocol

[9]Utilizing the recording interface, each participant chose a recipe from the selected 24 WikiHow recipes.

[10]The term *errors* is synonymous with what the AI community typically calls *mistakes* (cf. [19]).

[11]The practice of using scripted videos for activity understanding [82] has inspired us to develop the strategies.

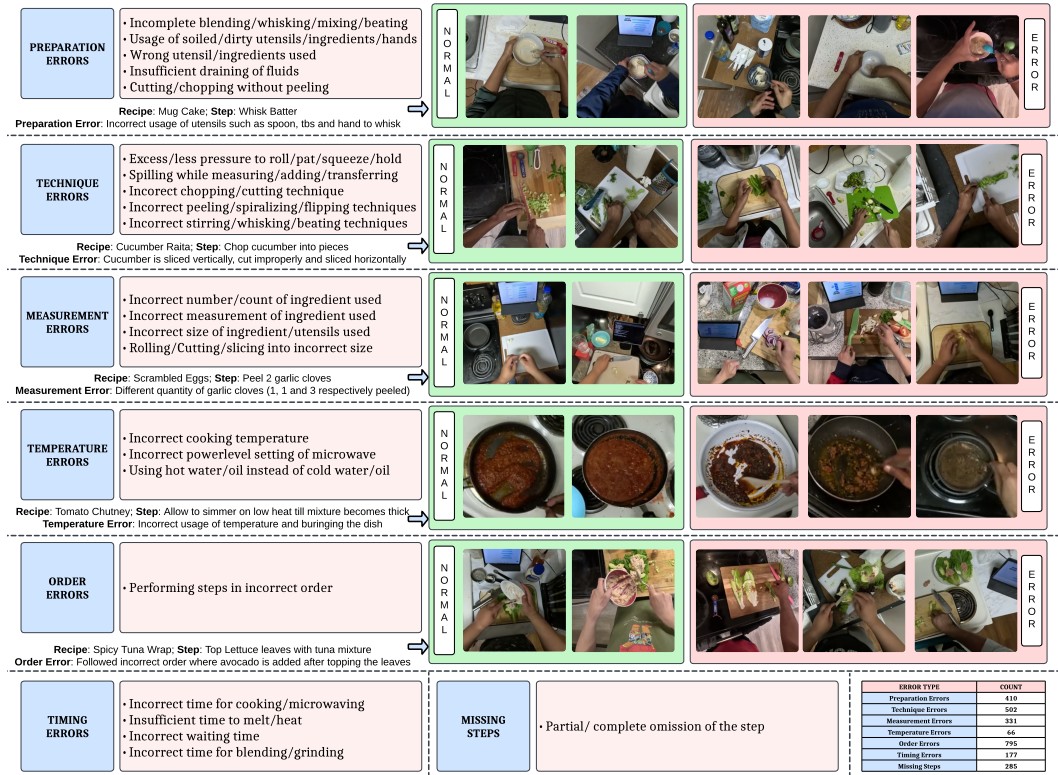

Figure 3: **Error Categories.** Left: We present a categorization of participant-induced errors derived from the annotated error descriptions of the recordings. Right: We display frames captured from various recordings, highlighting correct and erroneous executions. Bottom Right: We present statistics on the error categories in the dataset derived from the compiled annotations of all recordings.

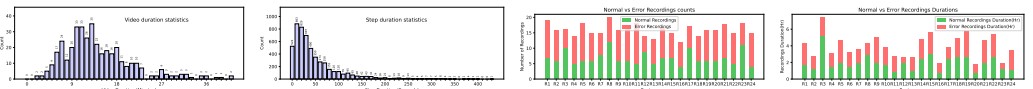

Figure 4: **Statistics.** Top: We present video and step duration statistics to the left & right respectively. Bottom: We present the total count and the durations of normal and error recordings for each recipe.

interface to create an error script for each selected recipe recording. The modified recipe steps were displayed on a tablet, enabling participants to perform according to their scripted errors. In Fig. 4, we present both general statistics of the dataset and specific statistics of normal and error recordings.

## 3.2 Data Annotation

We implemented a dual-layer review process to guarantee high-quality annotations. Each video was first annotated by the person who recorded it and then reviewed for accuracy by a second reviewer. Thus ensuring that all errors were correctly captured in the annotations corresponding to each step. Our annotations are structured to provide detailed insights into the recorded actions, facilitating both coarse-grained and fine-grained action analyses. Specifically, we offer the following annotations: (1) **Coarse-Grained Actions:** We mark the start and end times of each step in a recording of the recipe. (2) **Fine-Grained Actions:** For 20% of our data, we provide fine-grained action annotations to support semi/weakly supervised learning techniques for action recognition. (3) **Error Descriptions:** For each step, if an error occurs during its execution, we link its step annotation with the specific category of the error and a description of the error, thus enabling a comprehensive understanding.

**Coarse-Grained Action/Step Annotations.** We designed an interface for annotating steps in Label Studio[12]. Annotators are presented with this interface to mark each step's start and end times. Our coarse-grained actions/steps are significantly longer than a single fine-grained action and encompass multiple such fine-grained actions to perform the described step successfully. For example, to

---

[12]https://labelstud.io/

accomplish the step *{Chop a tomato}*, we include the following in the annotation (1) **Pre-conditional actions:** *{opening refrigerator, grabbing a polythene bag of tomatoes, taking a tomato, placing the tomato on cutting board, close fridge}* (2) **Post-conditional actions:** *{placing down the knife, grabbing the polythene bag of tomatoes, opening refrigerator and placing the bag in the refrigerator}*.

**Fine-Grained Action Annotations.** Inspired by the pause-and-talk [11], we have developed a web-based tool for fine-grained action annotations using Whisper [74] (for speech-to-text translation).

**Error Category Annotations.** Following each recording, participants were also asked to categorize errors performed in each step based on a set of guidelines. Specifically, we ask participants to broadly classify an error as a (1) *Preparation Error* when they use soiled/wrong ingredients or use different tools, (2) *Measurement Error* when they use wrongly measured ingredients, (3) *Timing Error* when they perform a step in shorter or longer duration than what is prescribed (e.g. Microwave for Microwave instead of 30 seconds) (4) *Temperature Error* when they set higher/lower power levels in the microwave or on a stove than what is prescribed (5) *Missing Step* when they omit to perform a step (6) *Technique Error* when they perform the required action incorrectly, leading to a wrong outcome than expected. (7) *Order Error* when they execute steps out of the required sequence. We compile and present the categorization of errors, their descriptions and visual illustrations in Fig. 3.

# 4 Experiments

Our experiments are designed to address the following questions: (Q1) What is the efficacy of transfer learning in recognizing errors? (Q2) How effective are current Vision Language Models (VLMs) in zero-shot error recognition? (Q3) How do state-of-the-art Multi-Step Localization methods perform on our dataset, particularly in terms of robustness to technique errors? (Q4) How do current self-supervised procedure learning methods in literature perform when applied to our dataset[13]?

**Features.** To answer the above questions we trained[14] our proposed baseline models on features obtained using pre-trained models such as 3D-ResNet [34], SlowFast [24], X3D [23], VideoMAE [91], Imagebind [30] and Omnivore [31] which were originally trained for video recognition tasks. Specifically, we split each video into 1-second sub-segments and extracted features to train models.

## 4.1 Error Recognition

This section answers questions (Q1) and (Q2); specifically, we address **Q1** by formulating the error recognition task as a *supervised binary classification* problem. We proposed three architectural variants (Fig. 5) as our baseline models and trained them using video/multimodal features. To address **Q2**, we employed a *prompt-and-predict* paradigm to recognize errors in activity recordings. Specifically, we formulated the problem as a Video Question Answering task (Fig. 6), crafted targeted question prompts using task graphs and error annotations(Fig.3); supplied these engineered prompts along with the videos as input to a VLM and evaluated its performance in zero-shot error recognition.

**Supervised Error Recognition (SupervisedER).** We utilized the features extracted using pre-trained models to train variants of our baseline binary classification models and evaluated trained models using the standard metrics such as accuracy, precision, recall, F1 and AUC (see Table 2). Specifically, we trained our models to classify each step of a video into one of two classes *{error(1), normal(0)}*. We constructed two data splits, step and recording splits, for training error recognition models. For the step split, we first compiled a dataset of video segments corresponding to all steps of all recipes in the dataset. Then divided it into train, validation, and test subsets. For the recordings split, we compiled all the recordings of all recipes in the dataset and divided the dataset into train, validation, and test subsets. Using error annotations (Figure 3), we first generated class labels for all video segments corresponding to the recipe steps. Then, we assigned the class label corresponding to the step to all 1-second sub-segments within the step and trained baseline models. During inference, we assigned the majority class label of the sub-segments corresponding to a step as the label to that step.

We proposed three architectural variants as baselines:$\{\mathcal{V}_1, \mathcal{V}_2, \mathcal{V}_3\}$ (see Fig. 5). In $\mathcal{V}_1$, we used the extracted features and constructed labels as described above and trained a Multi-Layer Perceptron (MLP) head. This approach assesses the efficacy of visual cues identified by pre-trained video

---

[13]Characterized by longer step durations, refer App. C for a comparison across procedural activity datasets

[14]We present the details about hyperparameters used for training the proposed baseline models in Appendix B.

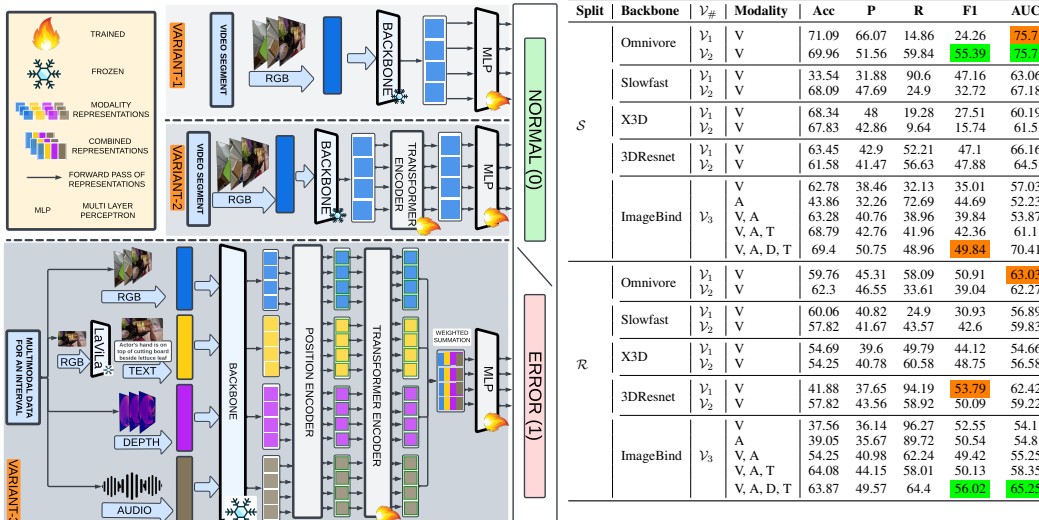

Figure 5: **SupervisedER** architectures of 3 baselines.

| Split | Backbone | $\mathcal{V}_\#$ | Modality | Acc | P | R | F1 | AUC |
|---|---|---|---|---|---|---|---|---|
| $\mathcal{S}$ | Omnivore | $\mathcal{V}_1$ | V | 71.09 | 66.07 | 14.86 | 24.26 | 75.7 |
| | | $\mathcal{V}_2$ | V | 69.96 | 51.56 | 59.84 | 55.39 | 75.7 |
| | Slowfast | $\mathcal{V}_1$ | V | 33.54 | 31.88 | 90.6 | 47.16 | 63.06 |
| | | $\mathcal{V}_2$ | V | 68.09 | 47.69 | 24.9 | 32.72 | 67.18 |
| | X3D | $\mathcal{V}_1$ | V | 68.34 | 48 | 19.28 | 27.51 | 60.19 |
| | | $\mathcal{V}_2$ | V | 67.83 | 42.86 | 9.64 | 15.74 | 61.5 |
| | 3DResnet | $\mathcal{V}_1$ | V | 63.45 | 42.9 | 52.21 | 47.1 | 66.16 |
| | | $\mathcal{V}_2$ | V | 61.58 | 41.47 | 56.63 | 47.88 | 64.5 |
| | ImageBind | $\mathcal{V}_3$ | V | 62.78 | 38.46 | 32.13 | 35.01 | 57.03 |
| | | | A | 43.86 | 32.26 | 72.69 | 44.69 | 52.23 |
| | | | V, A | 63.28 | 40.76 | 38.96 | 39.84 | 53.87 |
| | | | V, A, T | 68.79 | 42.76 | 41.96 | 42.36 | 61.1 |
| | | | V, A, D, T | 69.4 | 50.75 | 48.96 | 49.84 | 70.41 |
| $\mathcal{R}$ | Omnivore | $\mathcal{V}_1$ | V | 59.76 | 45.31 | 58.09 | 50.91 | 63.03 |
| | | $\mathcal{V}_2$ | V | 62.3 | 46.55 | 33.61 | 39.04 | 62.27 |
| | Slowfast | $\mathcal{V}_1$ | V | 60.06 | 40.82 | 24.9 | 30.93 | 56.89 |
| | | $\mathcal{V}_2$ | V | 57.82 | 41.67 | 43.57 | 42.6 | 59.83 |
| | X3D | $\mathcal{V}_1$ | V | 54.69 | 39.6 | 49.79 | 44.12 | 54.66 |
| | | $\mathcal{V}_2$ | V | 54.25 | 40.78 | 60.58 | 48.75 | 56.58 |
| | 3DResnet | $\mathcal{V}_1$ | V | 41.88 | 37.65 | 94.19 | 53.79 | 62.42 |
| | | $\mathcal{V}_2$ | V | 57.82 | 43.56 | 58.92 | 50.09 | 59.22 |
| | ImageBind | $\mathcal{V}_3$ | V | 37.56 | 36.14 | 96.27 | 52.55 | 54.1 |
| | | | A | 39.05 | 35.67 | 89.72 | 50.54 | 54.8 |
| | | | V, A | 54.25 | 40.98 | 62.24 | 49.42 | 55.25 |
| | | | V, A, T | 64.08 | 44.15 | 58.01 | 50.13 | 58.35 |
| | | | V, A, D, T | 63.87 | 49.57 | 64.4 | 56.02 | 65.25 |

Table 2: **SupervisedER** evaluation results of baselines. Variant type ($\mathcal{V}_\#$), Modality of features (M), Video (V), Audio (A), Depth (D), and Text (T).

recognition models in recognizing errors in sub-segments. In $\mathcal{V}_2$, we shifted our focus from sub-segment prediction and trained a transformer that processed all video sub-segments corresponding to each step. This method is designed to capitalize on the long-term temporal cues present within the video segments of recipe steps to enhance the prediction performance of the trained models. In variant $\mathcal{V}_3$, we harnessed the multimodal data of recordings and trained a unified transformer model. This approach employed an attention mechanism to integrate information from all modalities of data.

**Insights.** Our $\mathcal{V}_2$ models consistently outperformed $\mathcal{V}_1$ models. Incorporating additional data modalities like audio, text, and depth into our $\mathcal{V}_3$ models significantly improved their performance. Our omnivore-based $\mathcal{V}_2$ models performed similarly to our $\mathcal{V}_3$ models trained using Imagebind[15] features.

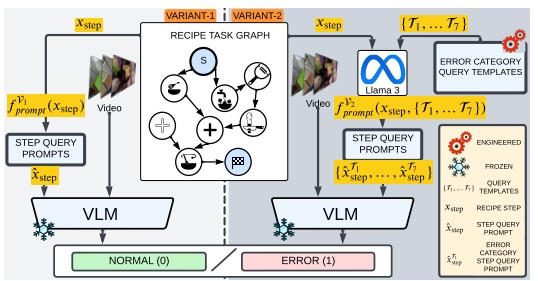

Figure 6: **ZeroShotER** evaluation pipeline of VLMs

| VLM | Variant | Acc | P | R | F1 |
|---|---|---|---|---|---|
| Video-LLaVa [55] | $\mathcal{V}_1$ | 64.3 | 34.2 | 3.9 | 6.7 |
| | $\mathcal{V}_2$ | 52.85 | 36.3 | 49.3 | 41.8 |
| TimeChat [76] | $\mathcal{V}_1$ | 65.0 | 51.11 | 1.15 | 2.26 |
| | $\mathcal{V}_2$ | 43.5 | 34.38 | 69.7 | 46.1 |

Table 3: **ZeroShotER** evaluation results.

**Zero-Shot Error Recognition (ZeroShotER).** We proposed two baseline variants, $\mathcal{V}_1$ and $\mathcal{V}_2$, for ZeroShotER[16] using prompt-and-predict architectures, as illustrated in Fig. 6. Both variants involve constructing query prompts for each recipe step and querying VLMs using these prompts along with video inputs. We utilized state-of-the-art[17] open-source VLMs, Video-LLaVa [55] and TimeChat [76], to recognize errors and reported the evaluation results using standard metrics, such as accuracy, precision, recall, and F1 scores. Specifically, for $\mathcal{V}_1$, we leveraged the associated task graphs of each recipe and generated a prefix question prompt for each step. These prompts were then used to query VLMs to recognize errors in videos. We employed a prompt-ensembling approach in $\mathcal{V}_2$ to recognize errors in a shift from the single-prompt strategy. Specifically, we designed prompt templates tailored to each error category (refer to Appendix B for examples). Using Llama3 [92], we generated a set

---

[15]We chose Imagebind to represent visual, textual, and auditory data in a unified embedding space.

[16]We also adapted anomaly detection methods for zero-shot error recognition (refer to Appendix B.)

[17]State-of-the-art at the time of writing this paper

of query prompts tailored to specific recipe steps and error categories. We combined the VLMs' predictions for error category-specific queries using an OR operation to determine the prediction for each step. This strategy of using error category-specific queries focused the VLMs' attention on specific segments of the video and enhanced the overall performance as illustrated in Table. 3.

**Insights.** While VLMs are adept at interpreting short video segments and answering straightforward questions (Eg. *Is there cucumber in the video?, Is the person peeling the cucumber?*), they struggle with tasks that require assimilation of information from various time intervals in long videos. This limitation becomes evident in tasks like error recognition in ego-centric videos, where questions such as *"Did the person chop or grate a completely peeled cucumber?"* require a deeper understanding.

**Remark.** The low scores indicate the complexity of the above tasks and call for developing sophisticated methods that semantically understand the context, meaning, and cause of various errors.

## 4.2 Multi Step Localization (MSL)

In supervised MSL, we aim to simultaneously identify the start and end boundaries of steps within an untrimmed long video and classify these identified video segments. This section addresses the question (Q3) by splitting it into two parts. For the first part—*How do state-of-the-art supervised MSL methods fare on our dataset?*; We utilized the features extracted from pre-trained video recognition models and trained our baseline models which employed ActionFormer [101] head on three proposed splits (indicated using the symbols $\{\mathcal{E}, \mathcal{P}, \mathcal{R}\}$ and detailed in App.A) of the original dataset. For the second part - *How robust are MSL methods to technique errors?*; We trained baseline models on three proposed splits ($\{\mathcal{E}, \mathcal{P}, \mathcal{R}\}$) of a modified dataset where the training subset exclusively contained normal recordings without errors. We then evaluated these models on two distinct test sets that exclusively comprised either normal recordings or error recordings (indicated using the symbols $\mathcal{T}_n$ and $\mathcal{T}_e$ respectively). We reported results using the standard MSL/TAL metrics such as Recall@K (R@K) and Mean Average Precision (mAP) across different temporal Intersections over Unions ($\mathcal{I}_t$). We presented our evaluation for both the tasks, namely, Multi-Step Localization (MSL) and Robust Multi-Step Localization (RobustMSL), in Tables 4 and 5 respectively. We also showcased qualitative examples of step localization for sampled normal and error recordings from 4 recipes in Figure 7.

Table 4: **MSL** evaluation results.

| $\mathcal{B}$ | $\mathcal{D}$ | $\mathcal{I}_t = 0.1$ | | | $\mathcal{I}_t = 0.3$ | | | $\mathcal{I}_t = 0.5$ | | |
|---|---|---|---|---|---|---|---|---|---|---|
| | | mAP | R@1 | R@5 | mAP | R@1 | R@5 | mAP | R@1 | R@5 |
| **3D Resnet** | $\mathcal{E}$ | 25.98 | 54.82 | 77.59 | 23.75 | 48.38 | 72.19 | 19.59 | 38.44 | 61.87 |
| | $\mathcal{P}$ | 29.29 | 63.07 | 88.96 | 27.71 | 56.60 | 84.75 | 23.21 | 46.79 | 76.86 |
| | $\mathcal{R}$ | 29.39 | 61.14 | 85.41 | 27.89 | 55.82 | 82.17 | 23.97 | 46.54 | 73.29 |
| **Slowfast** | $\mathcal{E}$ | 27.68 | 55.73 | 77.45 | 25.51 | 48.98 | 70.90 | 21.09 | 37.82 | 60.58 |
| | $\mathcal{P}$ | 32.77 | 63.22 | 90.43 | 31.21 | 58.82 | 86.82 | 27.25 | 50.70 | 79.49 |
| | $\mathcal{R}$ | 32.90 | 63.97 | 89.29 | 31.47 | 59.26 | 85.32 | 27.89 | 51.62 | 77.27 |
| **VideoMAE** | $\mathcal{E}$ | 28.12 | 51.76 | 73.00 | 26.38 | 46.16 | 67.87 | 21.35 | 37.12 | 57.81 |
| | $\mathcal{P}$ | 38.86 | 64.86 | 84.05 | 37.41 | 60.32 | 80.63 | 32.24 | 51.46 | 71.88 |
| | $\mathcal{R}$ | 37.44 | 63.08 | 80.90 | 35.11 | 57.30 | 77.38 | 30.76 | 49.19 | 69.43 |
| **Omnivore** | $\mathcal{E}$ | 40.40 | 67.51 | 87.69 | 38.32 | 62.31 | 82.82 | 33.41 | 53.01 | 72.85 |
| | $\mathcal{P}$ | 48.16 | 75.96 | 93.41 | 45.82 | 70.34 | 90.51 | 41.16 | 62.00 | 84.73 |
| | $\mathcal{R}$ | 44.81 | 73.71 | 93.34 | 42.76 | 68.14 | 89.82 | 37.19 | 56.93 | 81.86 |

Table 5: **RobustMSL** evaluation results.

| $\mathcal{B}$ | $\mathcal{D}$ | $\mathcal{T}$ | $\mathcal{I}_t = 0.1$ | | | $\mathcal{I}_t = 0.3$ | | | $\mathcal{I}_t = 0.5$ | | |
|---|---|---|---|---|---|---|---|---|---|---|---|
| | | | mAP | R@1 | R@5 | mAP | R@1 | R@5 | mAP | R@1 | R@5 |
| **VideoMAE** | $\mathcal{E}$ | $\mathcal{T}_n$ | 24.44 | 38.22 | 52.48 | 22.97 | 34.77 | 49.51 | 18.67 | 28.57 | 42.68 |
| | | $\mathcal{T}_e$ | 7.53 | 13.54 | 20.52 | 6.93 | 11.4 | 18.36 | 5.63 | 8.55 | 15.13 |
| | $\mathcal{P}$ | $\mathcal{T}_n$ | 26.78 | 37.43 | 46.28 | 25.68 | 34.79 | 44.6 | 22.02 | 29.43 | 39.81 |
| | | $\mathcal{T}_e$ | 16.98 | 27.43 | 37.76 | 16.46 | 25.53 | 36.03 | 14.64 | 22.03 | 32.07 |
| | $\mathcal{R}$ | $\mathcal{T}_n$ | 26.27 | 37.15 | 46.93 | 24.71 | 34.06 | 45.03 | 21.51 | 29.36 | 40.44 |
| | | $\mathcal{T}_e$ | 15.43 | 25.94 | 33.97 | 14.44 | 23.23 | 32.35 | 12.96 | 19.83 | 28.99 |
| **Omnivore** | $\mathcal{E}$ | $\mathcal{T}_n$ | 34.65 | 47.91 | 60.63 | 33.06 | 44.77 | 58.36 | 28.59 | 38.38 | 51.9 |
| | | $\mathcal{T}_e$ | 12.51 | 19.6 | 27.06 | 11.66 | 17.54 | 24.45 | 9.94 | 14.63 | 20.96 |
| | $\mathcal{P}$ | $\mathcal{T}_n$ | 32.5 | 44.45 | 52.47 | 31.13 | 41.53 | 50.91 | 28.39 | 37.03 | 47.97 |
| | | $\mathcal{T}_e$ | 21.28 | 31.51 | 40.93 | 20.12 | 28.81 | 39.6 | 18.08 | 24.96 | 36.77 |
| | $\mathcal{R}$ | $\mathcal{T}_n$ | 30.22 | 42.43 | 52.11 | 28.94 | 39.47 | 50.49 | 25.15 | 32.65 | 46.51 |
| | | $\mathcal{T}_e$ | 19.54 | 31.28 | 41.24 | 18.4 | 28.66 | 39.33 | 16.27 | 24.28 | 35.35 |

Figure 7: **MSL** qualitative results of the Omnivore-based model trained using the split $\mathcal{R}$.

**Insights.** We observed that our models based on Omnivore features consistently outperformed others. We also noticed that models trained using features extracted from longer video sub-segments (>1sec) outperformed those trained using features extracted from 1-sec video sub-segments (App.B). However, all our trained models using the modified dataset exhibited poor performance on $\mathcal{T}_e$ compared to $\mathcal{T}_n$.

**Remark.** We conjecture that exploiting semantic information from task graphs and employing probabilistic filtering methods such as particle filters to refine predictions could enhance performance.

## 4.3 Procedure Learning

Given long, untrimmed videos of procedural activities where the sequences of steps can be performed in multiple orders, self-supervised procedure learning methods aim to identify relevant frames across videos of an activity and estimate the sequential steps required to complete the activity. In this section, we address the question **Q4** by simultaneously answering *(a) Can we infer the underlying procedure (recipe text) from the videos of a particular recipe and (b) How does the self-supervised procedure learning methods in literature perform on the proposed dataset?*. Specifically, we answered both parts by comparing the performance of our models trained on the proposed dataset using self-supervised procedure learning methods [3, 16] against the random setting defined by EgoProceL [3](see Tab. 6). We followed the setup described in EgoProceL and trained two embedder networks, one using the Cycleback Regression loss ($\mathcal{C}$) [$\mathcal{M}_1$] [16] and the other using a blend of two loss functions: Cycleback Regression loss ($\mathcal{C}$) and Contrastive - Inverse Difference Moment loss ($\mathscr{C}$) [$\mathcal{M}_2$][3]. The combined loss function is $\mathcal{C} + \lambda \times \mathscr{C}$, where $\lambda$ is a hyperparameter. While we exclusively used these loss functions to train the embedder networks, we continued using the Pro-Cut Module to categorize frames into key steps. We presented evaluation results for 5 recipes Tab. 6 and all recipes in App. B.

Table 6: **Procedure Learning.** Here, $\mathcal{P}$ represents precision, $\mathcal{R}$ represents recall, and $\mathcal{I}$ represents IOU.

| Recipe | Random | | | $\mathcal{M}_1$ [16] | | | $\mathcal{M}_2$ [3] | | |
|---|---|---|---|---|---|---|---|---|---|
| | $\mathcal{P}$ | $\mathcal{R}$ | $\mathcal{I}$ | $\mathcal{P}$ | $\mathcal{R}$ | $\mathcal{I}$ | $\mathcal{P}$ | $\mathcal{R}$ | $\mathcal{I}$ |
| BlenderBananaPancakes | 7.40 | 3.83 | 2.26 | 12.65 | 9.50 | 5.16 | 15.54 | 9.96 | 5.72 |
| Coffee | 6.54 | 3.87 | 2.17 | 13.68 | 9.91 | 5.49 | 15.76 | 10.25 | 5.63 |
| MugCake | 5.45 | 4.00 | 2.12 | 16.12 | 12.95 | 6.87 | 10.32 | 8.85 | 4.40 |
| PanFriedTofu | 5.35 | 3.97 | 1.54 | 8.86 | 10.39 | 3.75 | 9.34 | 12.44 | 3.87 |
| Pinwheels | 6.54 | 4.28 | 2.13 | 13.58 | 11.96 | 5.92 | 16.08 | 13.06 | 7.05 |
| **Average of 24 recipes** | **7.61** | **3.92** | **2.22** | **15.62** | **10.85** | **5.78** | **15.78** | **10.68** | **5.82** |

**Insights.** Our models significantly outperformed the predefined random setting, demonstrating the feasibility of inferring procedural steps from our dataset. However, these models scored lower on our dataset compared to existing procedure learning datasets. We believe this drop in performance is mainly due to our dataset's unique challenge, which includes videos with longer key step durations.

**Additional Results.** We provide several analyses in Appendix B, including (a) Error Category Recognition, (b) Early Error Recognition, (c) Anomaly Detection and (d) Ablation studies for MSL.

## 5 Discussion, Limitations and Future Work

**Discussion.** We introduced a novel egocentric dataset for understanding errors in procedural activities. Our dataset consists of synchronized egocentric views, audio, and depth information specifically designed for tasks such as 3D activity analysis, Procedure Learning, Error Recognition, and more. While current methods have yielded promising outcomes, they continue to struggle to tackle these challenges adequately with satisfactory results, as demonstrated by our experimental assessment. This indicates the need for further exploration in this domain.

**Limitations.** We aimed to capture deviations during procedural activities from an egocentric perspective. Since such data cannot be sourced from crowd-sourced platforms, we captured participant data while performing procedural activities. By the nature of the problem, errors that occur when performing procedural activities are combinatorial and can have a compounding effect. Thus, our work has the following limitations: (1) For each activity, the errors captured and presented in the dataset form a subset of the whole combinatorial space; (2) Capturing 4D data in real kitchen environments posed logistical and equipment training challenges. As a result, we were compelled to limit the data collection to a specific geographic area.

**Future Work.** Our work opens up several avenues for future work. First, an exciting direction is the extension of the dataset to include activities from other domains. By incorporating tasks such as executing hardware-related activities (e.g., working with cars or computer parts), the dataset can encompass a wider range of activities. Second, the dataset can be used to compare and develop methods for solving various tasks such as Few-Shot Error Recognition using visual/textual prompts, Semantic Role Labelling, Long Video Understanding, Procedure Planning, Reducing Errors, etc.

## Acknowledgements

This work was supported in part by the DARPA Perceptually-Enabled Task Guidance (PTG) Program under contract number HR00112220005, by the DARPA Assured Neuro Symbolic Learning and Reasoning (ANSR) Program under contract number HR001122S0039, by the National Science Foundation grant IIS-1652835 and by the AFOSR award FA9550-23-1-0239. We would like to express our gratitude to all the reviewers for their insightful comments and queries, which have substantially enhanced the quality of our manuscript. We would also like to extremely thank Sai Krishna, Zhang Yang, Jihan Wang, Enoch Chiu, Pratyush Kaware, Saurabh Bhole, Ankit Upadhyay, Vashist Gupta, Samuel, Jyothise, Shruthi and Manogna for allowing us to collect data in their kitchens.

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

# Appendices

# A  Overview

## A.1  Motivation

**Procedural Datasets.**    We present our motivation to collect a new dataset with errors. Current datasets that study procedural tasks, such as GTEA [21], Breakfast [51], CMU-MMAC [13], 50Salads [87], COIN [89], CrossTask [105], ProceL [17], EgoProceL[3], Assembly101 [19], and HowTo100M [60], encompass temporal variation in the order of the steps performed. However, these datasets are predominantly sourced from crowd-sourced online platforms, resulting in the videos often containing drastically different steps, with alterations impacting more than 30% of the content. Our interest lies in understanding errors induced by deviating from the given instruction set. To this end, we require two types of videos: normal ones that closely follow the instructions and error videos that depict deviations. Moreover, we aim to capture these videos from an ego-centric perspective to minimize the occlusions typical in third-person videos. We are primarily interested in understanding errors when the objects under the interaction continuously change shape and colour during a procedural activity.

**Recent Progress.**    Error recognition in procedural activities has received significant traction, leading to the proposal of new datasets with errors [93, 29, 37, 80]. Although they aim to identify errors in procedural activities, they focus on tasks related to assembly and disassembly. **The activities involve objects with constant shapes and colors, which lack the desired characteristics**. The absence of such specific video resources led us to curate a dataset (Fig. 8) embodying all our desired characteristics. By focusing on cooking activities with desired characteristics, our dataset can be used to develop easily transferable algorithms for other sensitive domains, such as medicine and chemistry.

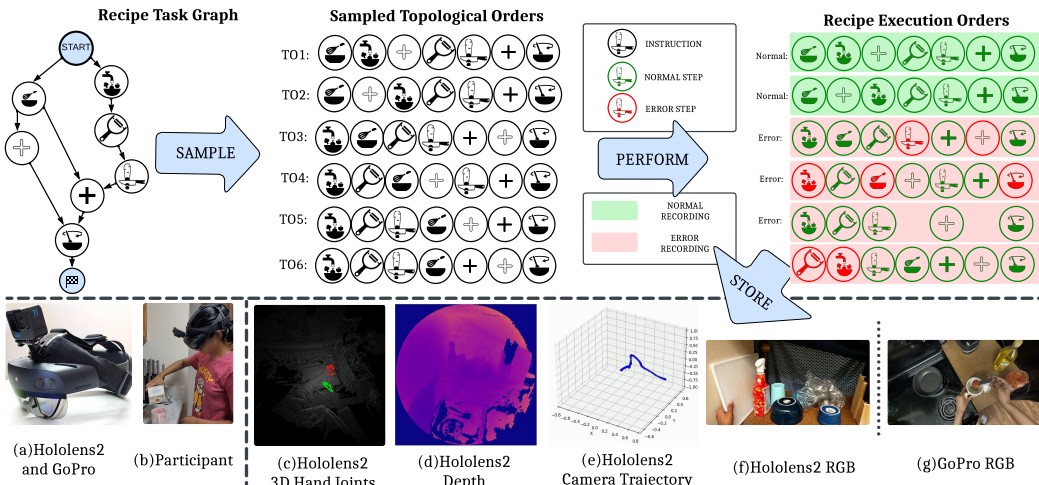

Figure 8: **Overview.** Top: We constructed task graphs for the selected recipes. These graphs facilitated sampling partial orders (cooking steps) that participants followed to perform. During the execution of some of these steps, participants induced errors that are both **intentional** and **unintentional** in nature. Bottom: On the left, we present the sensors employed for data collection, and on the right, we describe the details of the modalities of the data collected while the participant performs the recipe.

In the following sections, we will start by explaining how to use the data, including details about the structured splits of the dataset. We will then discuss comprehensive results and various analyses of the tasks we have benchmarked. Lastly, we will describe our data collection and annotation processes involving three stages: (a) Data collection planning, (b) Data collection, and (c) Data processing.

## A.2 Data Splits

We created diverse splits for training models on the proposed dataset where each split is based on a specific criteria ensuring diversity in the train, validation and test subsets according to it. This approach enables models to concentrate on different facets of the data. The splits are categorized as follows: (1) Recording Environment ($\mathcal{E}$), (2) Recording Person ($\mathcal{P}$), (3) Recipes ($\mathcal{R}_e$), (4) Recordings ($\mathcal{R}$), (5) Steps ($\mathcal{S}$), and (6) Recording Type ($\mathcal{R}_t$).

**(1) Environment ($\mathcal{E}$).** Our dataset comprises data collected from ten different environments, with a larger proportion of recordings sourced from five of these environments. We used this information to strategically divide the dataset. Recordings from these five environments were included in both the training and validation sets, while recordings from the remaining environments were allocated to the test set. We ensured a consistent balance of normal and error recordings across all three sets.

**(2) Persons ($\mathcal{P}$).** Eight participants compiled our dataset, each recording an equal number of videos. To facilitate a balanced distribution, we designed a split that includes recordings from two participants—who performed all the recipes—in the test set. The recordings from the remaining participants were divided between the training and validation sets.

**(3) Recipes ($\mathcal{R}_e$).** We meticulously divided 24 selected recipes into training, validation, and test sets based on the specific skills required for each recipe. By identifying all the essential skills needed to execute these recipes, we ensured that each set included recipes that necessitate applying these skills. This strategic division facilitates learning tasks that involve skill transfer.

**(4) Recordings ($\mathcal{R}$).** We categorize all recordings of a recipe into training, validation, and test sets according to a specified ratio. This split is generated randomly and varies with each iteration.

**(5) Steps ($\mathcal{S}$).** We compile a comprehensive dataset consisting of video segments that correspond to the steps of all recordings. This dataset is then divided into training, validation, and test splits, ensuring that steps from each recording are represented across all three splits.

**(6) Recording Type ($\mathcal{R}_t$).** Tasks that require a semantic understanding of errors employ methods that differentiate between normal and error recordings. The models can be trained using only normal recordings to learn the baseline behaviour and then applied to recognize errors in recordings.

# B    Benchmarking

We present comprehensive evaluation results and analyses for our proposed dataset on several tasks:

1. **Error Recognition**: This includes evaluations under two different settings:
   - **Zero-Shot**:
     - Error Recognition
     - Error Category Recognition
     - Anomaly Detection
   - **Supervised**:
     - Error Recognition
     - Early Error Recognition
2. **Multi-Step Localization**
3. **Procedure Learning**

## B.1    Zero-Shot Error & Error Category Recognition

Error Recognition demands an accurate interpretation of actions and their consequences. This entails developing models that can semantically understand the progression of events during an activity and also assess the quality of the actions observed. Recently, Vision-Language Models (VLMs) have shown great promise in visual reasoning by combining effective visual analysis with the strong common-sense reasoning abilities of Large Language Models (LLMs). Therefore, our goal is to test the ability of recently proposed VLMs to recognize errors in video recordings of procedural activities.

Leveraging the prompt-and-predict paradigm we proposed two variants[18] $\{\mathcal{V}_1, \mathcal{V}_2\}$ for error recognition. We set up the Zero-Shot Error Recognition task as a Video Question Answering (VQA) problem. Our proposed variants $\{\mathcal{V}_1, \mathcal{V}_2\}$ primarily focused on the generation of task-specific questions, while relying on the existing state-of-the-art pre-trained VLMs for visual interpretation and the reasoning ability required (specific to visual interpretation) to answer the generated task-specific questions.

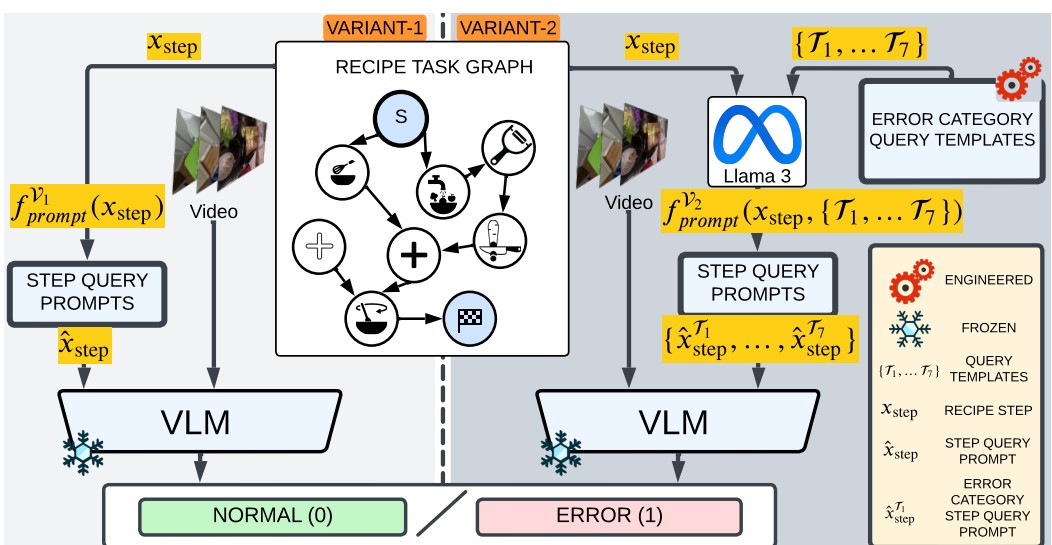

Figure 9: **ZeroShotER** evaluation pipeline of VLMs

Our proposed variants are distinguished by their use of single-prompt and multi-prompt approaches, as defined in the literature on prompt engineering. Specifically, in $\mathcal{V}_1$, we leverage the task graphs and error descriptions provided as part of annotations to construct questions (a single question prompt specific to each step of the recipe) that enquire about the completion of a recipe step in the

---

[18]We only probe the textual inputs, leaving the visual interpretation aspects of the VLMs unchanged.

procedural activity videos. In $\mathcal{V}_2$, instead of a single-prompt (more general questions), we adopt a prompt-ensembling strategy (more specific questions) to recognize errors that occur in recordings.

Our $\mathcal{V}_2$ can be understood as follows: Error Recognition as a task requires identification of all errors that occur in procedural activity videos. Since the space of the possible errors that can potentially occur for each procedural activity is very large and combinatorial in nature, tasking a VLM to enquire about all possible errors through more general questions leads to significantly low performance (can be observed through numbers of $\mathcal{V}_1$ in Table 7). To address this, as illustrated in Figure 9, we leveraged the structured knowledge about the categories of errors that can occur while executing a procedural activity and crafted targeted question prompts. This strategy not only guides VLMs to answer more specific questions, thereby improving the performance scores (refer Table 7) but also aids in developing a systematic framework for building error recognition models using VLMs.

| VLM | Variant | Acc | P | R | F1 |
|---|---|---|---|---|---|
| Video-LLaVa [55] | $\mathcal{V}_1$ | 64.3 | 34.2 | 3.9 | 6.7 |
| | $\mathcal{V}_2$ | 52.85 | 36.3 | 49.3 | 41.8 |
| TimeChat [76] | $\mathcal{V}_1$ | 65.0 | 51.11 | 1.15 | 2.26 |
| | $\mathcal{V}_2$ | 43.5 | 34.38 | 69.7 | 46.1 |

Table 7: **ZeroShotER** evaluation results.

In subsequent sections, we detail the two proposed variants, $\{\mathcal{V}_1, \mathcal{V}_2\}$, including examples of the crafted question prompts for $\{\mathcal{V}_1, \mathcal{V}_2\}$. We also present our evaluation results (using standard binary classification metrics) for the tasks of Error Recognition and Error Category Recognition. We note that although we present evaluation results for two open-source VLMs, Video-LLaVa and TimeChat, our framework can be easily extended to closed-source VLMs such as GPT-4V and GeminiPro.

### B.1.1 Variant-1 ($\mathcal{V}_1$):

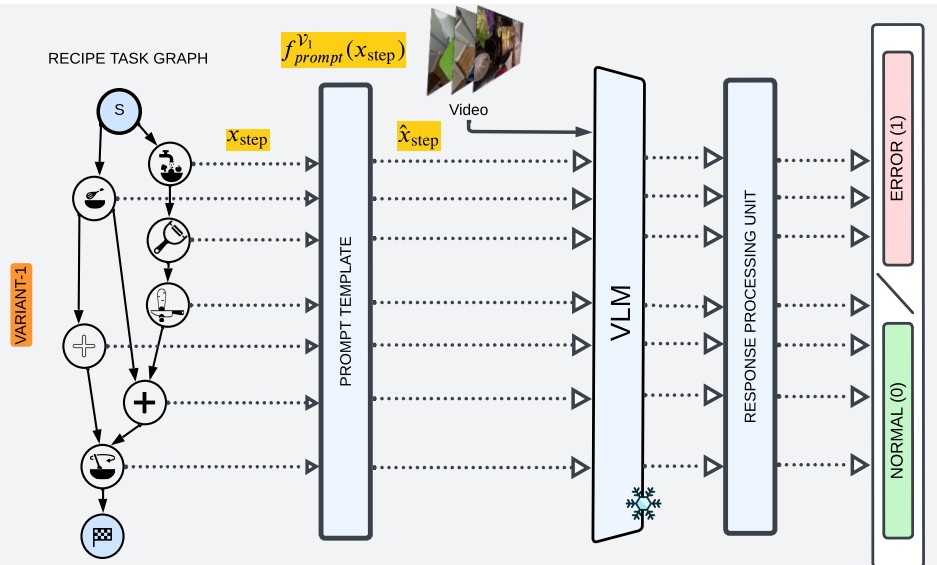

Figure 10: **ZeroShotERV1.** We outline the methodology for the proposed variant $\mathcal{V}_1$ as follows: Initially, we integrated the raw textual descriptions of steps extracted from task graphs into an engineered prompt template to create a single-question prompt specific to each step of the recipe. This prompt, along with the corresponding video, is inputted into a VLM. We analyze the responses generated from the VLM to obtain its predictions corresponding to each step of the recipe.

**Task.**    In Figure 10, we present overview of the proposed method. The two important components of $\mathcal{V}_1$ include (a) An engineered prompt template that is specific to the chosen VLM and (b) A response processing unit. For engineering the prompt template, we employed the following methodology:

- Inspired by the *Chain of Thought* prompting strategy, We formulated a list of templates that can potentially be used for recognizing errors in procedural activity recordings.
- We selected a diverse set of videos representing every recipe type included in the dataset.
- We executed our proposed pipeline with all the prompt templates on a selected set of videos and chose the best-performing prompt template corresponding to each VLM.

We noticed that although we craft the template to generate the response in a specific format, the response generated by VLM often follows a different format; thus, we included a response processing unit to convert the generated response into a preferred format. We presented results in Table 7.

In the subsequent sections, we present the engineered templates (tailored to the specific choice of VLMs) used to construct the question prompts for each step followed by a few examples of the steps sampled from the recipes included in the proposed dataset. We built our candidates for the templates utilizing the examples provided by the authors of employed VLMs, Video-LLaVa and TimeChat.

**Prompt Template.** Following are the engineered prompt templates corresponding to the VLMs

- **Video-LLaVa:** ASSISTANT: {Did, Is, Does, . . . } the person {perform, execute, doing, . . . } the step `recipe step` from the recipe `recipe` ?

- **TimeChat:** You are given a cooking video. Please watch the video and answer the following question: {Did, Is} the person {perform, doing} the step `recipe step` ? Return the answer in the format of Yes or No.

**Examples:**

---

Recipe: Cucumber Raita

Step: *In a mixing bowl, whisk 1 cup of chilled curd until smooth. Use fresh homemade or packaged curd*

VLM: Video-LLaVa

Prompt: Did the person whisk 1 cup of chilled fresh homemade or packaged curd until smooth while performing the Cucumber Raita recipe?

VLM: TimeChat

Prompt: You are given a cooking video. Please watch the video and answer the following question: Did the person whisk 1 cup of chilled fresh homemade or packaged curd until smooth? Return the answer in the format of Yes or No.

---

Recipe: Spiced Hot Chocolate

Step: *Add 2 pieces of chocolate to the mug*

VLM: Video-LLaVa

Prompt: Does the person add 2 pieces of chocolate to the mug while performing Spiced Hot Chocolate recipe?

VLM: TimeChat

Prompt: You are given a cooking video. Please watch the video and answer the following question: Does the person add 2 pieces of chocolate to the mug? Return the answer in the format of Yes or No.

---

Recipe: Tomato Mozzarella Salad

Step: *Rinse a tomato*

VLM: Video-LLaVa

Prompt: Did the person rinse one tomato?

VLM: TimeChat

Prompt: You are given a cooking video. Please watch the video and answer the following question: Did the person rinse one tomato? Return the answer in the format of Yes or No.

---

## B.1.2 Variant-2:

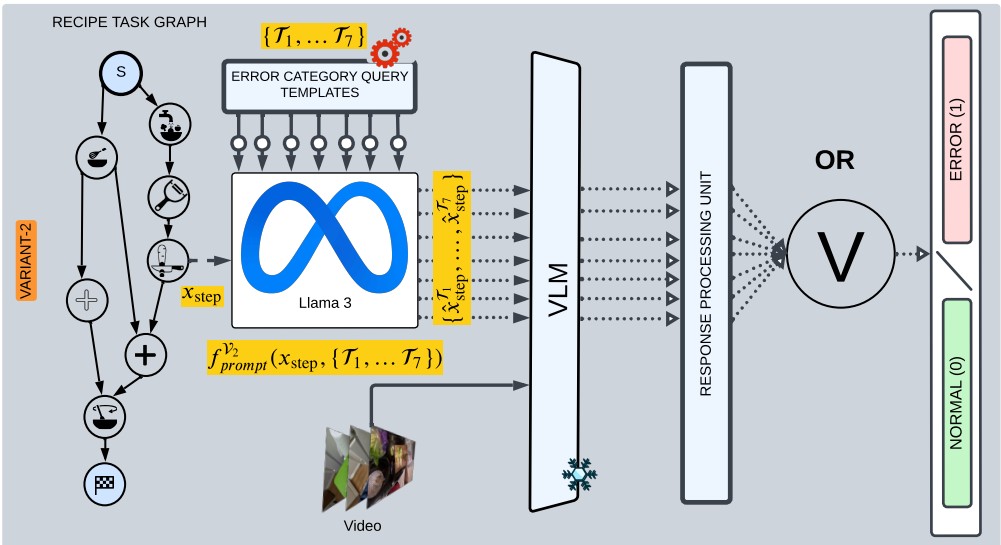

Figure 11: **ZeroShotERV2.** We outline the methodology for the proposed variant $\mathcal{V}_2$ as follows: Instead of a single-question prompt for each recipe step, we create seven question prompts, each tailored to a specific error category. Specifically, we engineered seven query templates, each corresponding to an error category. We fed these templates along with the description of each recipe step to Llama3 to generate seven tailored query prompts. We pass these error category-specific query prompts along with the videos to VLMs. We process the response generated by VLMs and apply an **OR** operation on processed responses of question prompts to obtain the final prediction for each step.

**Task.** In Figure 11, we present overview of the proposed variant $\mathcal{V}_2$. Important components of $\mathcal{V}_2$ are (a) Engineered error-category-specific query prompt templates tailored to VLM and (b) Final prediction generation. For engineering error-category-specific query prompt templates we followed:

– Inspired by the *Chain of Thought* prompting strategy, We formulated a list of templates that can be used for recognizing errors corresponding to *each error category* in recordings.

– We selected a diverse set of videos representing every recipe type included in the dataset.

– We executed our proposed pipeline using all error category-specific query prompt templates on a chosen set of videos. From these, we identified and selected the best-performing error category-specific query prompts for each VLM. We presented evaluation results in Table 7.

– We note that as an intermediate step, we also solve the Error Category Recognition task.

Table 8: **Error Category Recognition** evaluation results.

| Error Category | VLM | P | R | F1 | Acc |
|---|---|---|---|---|---|
| Order Error | Video-LLaVA | 16.3 | 35.3 | 22.3 | 65.5 |
| | TimeChat | 17.2 | 6.25 | 9.2 | 82.6 |
| Preparation Error | Video-LLaVA | 7.7 | 12.3 | 9.5 | 84.2 |
| | TimeChat | 7.1 | 50.1 | 12.4 | 52.2 |
| Measurement Error | Video-LLaVA | 0.0 | 0.0 | 0.0 | 93.7 |
| | TimeChat | 6.1 | 44.2 | 10.7 | 57.1 |
| Technique Error | Video-LLaVA | 11.1 | 0.4 | 0.7 | 90.9 |
| | TimeChat | 2.4 | 0.4 | 0.6 | 89.9 |
| Missing Steps | Video-LLaVA | 5.1 | 3.9 | 4.4 | 91.7 |
| | TimeChat | 2.2 | 0.4 | 12.4 | 94.3 |
| Temperature Error | Video-LLaVA | 0.5 | 4.6 | 0.9 | 89.4 |
| | TimeChat | 0.6 | 6.2 | 1.2 | 88.4 |
| Timing Error | Video-LLaVA | 6.6 | 0.6 | 1.1 | 96.8 |
| | TimeChat | 3.0 | 17.6 | 5.1 | 80.5 |

**Error Category Recognition.** In Table 8, we presented evaluation results for the task Error Category Recognition (namely, classify whether a video includes an error of a specific category or not.) formulated as a binary classification problem. We used the standard binary classification metrics to report the evaluation results. Specifically, we employed the following methodology:

- We construct an error category-specific question prompt for each step (refer Fig. 11).
- Using error annotations, we constructed error-category-specific label for each step.
- We processed the generated responses by VLM to obtain the predictions for recipe steps.
- We evaluated the obtained prediction using the labels constructed above. (refer Table 8).

Insights: (1) Video-LLaVa and TimeChat exhibit better performance on different categories of errors. Thus suggesting a VLM ensemble as a natural extension to estimate final predictions. (2) Error Category Recognition is a problem with heavy class imbalance, which can be inferred from the reported scores of accuracy and the F1 metrics in Tab. 8. (3) The low scores indicate the difficulty of the task, and we hope that these numbers will improve with more advanced closed-source VLMs.

**Examples:** We present category-specific templates and the corresponding examples.

---

**Error Category: Technique Error**

VLM: Video-LLaVa

Template: ASSISTANT: To prepare recipe , the person {should, has, ... } to {perform, execute, doing, ... } the step recipe step . Answer with a yes or no, {Did, Does, Has, ... } the person {carefully, precisely, ... } {perform, execute, doing, ... } the recipe step {without spilling, dropping, ... }?

VLM: TimeChat

Template: You are given a cooking video. Please watch the video and answer the following question: To prepare recipe , the person {should, has, ... } to {perform, execute, doing, ... } the step recipe step . {Did, Does, Has, ... } the person {carefully, precisely, ... } {perform, execute, doing, ... } the recipe step ? Return the answer in the format of Yes or No.

---

**Question Prompts**

---

Recipe: Cucumber Raita

Step: *Add the chopped or grated cucumber to the whisked curd.*

VLM: Video-LLaVa

Prompt: To prepare Cucumber Raita, the person has to add the chopped or grated cucumber to the whisked curd. Answer with a yes or no, does the person carefully add chopped or grated cucumber to the curd without spilling?

VLM: TimeChat

Prompt: You are given a cooking video. Please watch the video and answer the following question: To prepare Cucumber Raita, the person has to add chopped or grated cucumber to whisked curd. Does the person carefully add chopped or grated cucumber to the curd? Return the answer in the format of Yes or No.

---

Recipe: Spiced Hot Chocolate

Step: *Add 1/5 teaspoon cinnamon to the mug.*

VLM: Video-LLaVa

Prompt: To prepare Spiced Hot Chocolate, the person should add 1/5 teaspoon of cinnamon to the mug. Answer with a yes or no, did the person carefully add 1/5 teaspoon of cinnamon to the mug without spilling?

VLM: TimeChat

Prompt: You are given a cooking video. Please watch the video and answer the following question: To prepare Spiced Hot Chocolate, the person should add 1/5 teaspoon of cinnamon to the mug. Does the person carefully add 1/5 teaspoon of cinnamon to the mug without spilling? Return the answer in the format of Yes or No.

---

## Error Category: Preparation Error

| | |
|---|---|
| **VLM:** | Video-LLaVa |
| **Template:** | ASSISTANT: {What, Which, . . . } {tool, ingredient, . . . } is used for `recipe step` to make `recipe` ? {Select, Choose, . . . } one of the options: {option 1, option 2 . . . }. |
| | |
| **VLM:** | TimeChat |
| **Template:** | You are given a cooking video. Please watch the video and answer the following question: {What, Which, . . . } {tool, ingredient, . . . } is used for `recipe step` to make `recipe` ? {Select, Choose, . . . } one of the options: {option 1, option 2 . . . }. |

**Question Prompts**

| | |
|---|---|
| **Recipe:** | Cucumber Raita |
| **Step:** | *In a mixing bowl, whisk 1 cup of chilled curd until smooth. Use fresh homemade or packaged curd.* |
| **VLM:** | Video-LLaVa |
| **Prompt:** | What tool is used for making the chilled fresh homemade or packaged curd smooth in a mixing bowl for making Cucumber Raita? Choose one of the options: (a) whisker, (b) fork, (c) ladle, (d) knife. |
| **VLM:** | TimeChat |
| **Prompt:** | You are given a cooking video. Please watch the video and answer the following question: What tool is used for making the chilled fresh homemade or packaged curd smooth in a mixing bowl for making Cucumber Raita? Choose one of the options: (a) whisker, (b) fork, (c) ladle, (d) knife. |

| | |
|---|---|
| **Recipe:** | Spiced Hot Chocolate |
| **Step:** | *Microwave the contents of the mug for 1 minute.* |
| **VLM:** | Video-LLaVa |
| **Prompt:** | Which tool is used to heat the contents of the mug to make Spiced Hot Chocolate? Choose one of the options: (a) microwave, (b) saucepan, (c) toaster, (d) kettle. |
| **VLM:** | TimeChat |
| **Prompt:** | You are given a cooking video. Please watch the video and answer the following question: Which tool is used to heat the contents of the mug to make Spiced Hot Chocolate? Choose one of the options: (a) microwave, (b) saucepan, (c) toaster, (d) kettle. |

## Error Category: Order Error

**VLM:** Video-LLaVa

**Template:** ASSISTANT: {Did, Is, Does, ...} the person {perform, execute, doing, ...} the step `recipe step` to {cook, make} `recipe`? {Has, Have, ...} the `previous recipe step(s)` been {completed, performed, ...} before `recipe step`?

**VLM:** TimeChat

**Template:** You are given a cooking video. Please watch the video and answer the following question: {Did, Is, Does, ...} the person {perform, execute, doing, ...} the step `recipe step` to {cook, make} `recipe`? {Has, Have, ...} the `previous recipe step(s)` been {completed, performed, ...} before `recipe step`? Return the answer in the format of Yes or No.

**Question Prompts**

**Recipe:** Cucumber Raita

**Step:** *Peel the cucumber.*

**VLM:** Video-LLaVa

**Prompt:** Did the person peel the cucumber to make Cucumber Raita? Has 1 medium sized cucumber been rinsed before peeling the cucumber?

**VLM:** TimeChat

**Prompt:** You are given a cooking video. Please watch the video and answer the following question: Did the person peel the cucumber to make Cucumber Raita? Has 1 medium sized cucumber been rinsed before peeling the cucumber? Return the answer in the format of Yes or No.

**Recipe:** Spiced Hot Chocolate

**Step:** *Heat the contents of the mug for 1 minute and serve.*

**VLM:** Video-LLaVa

**Prompt:** Does the person heat the contents of the mug for 1 minute and served to cook Spiced Hot Chocolate? Have the contents of the mug been mixed before heating for 1 minute and serving?

**VLM:** TimeChat

**Prompt:** You are given a cooking video. Please watch the video and answer the following question: Does the person heat the contents of the mug for 1 minute and served to cook Spiced Hot Chocolate? Have the contents of the mug been mixed before heating for 1 minute and serving? Return the answer in the format of Yes or No.

**Error Category: Missing Steps**

VLM: Video-LLaVa

Template: {Did, Is, Does, ... } the person {perform, execute, doing, ... } the step `recipe step` from the recipe `recipe` ?

VLM: TimeChat

Template: You are given a cooking video. Please watch the video and answer the following question: {Did, Is} the person {perform, doing} the step `recipe step` ? Return the answer in the format of Yes or No.

**Question Prompts**

Recipe: Cucumber Raita

Step: *In a mixing bowl, whisk 1 cup of chilled curd until smooth. Use fresh homemade or packaged curd*

VLM: Video-LLaVa

Prompt: Did the person whisk 1 cup of chilled fresh homemade or packaged curd until smooth while performing the Cucumber Raita recipe?

VLM: TimeChat

Prompt: You are given a cooking video. Please watch the video and answer the following question: Did the person whisk 1 cup of chilled fresh homemade or packaged curd until smooth? Return the answer in the format of Yes or No.

Recipe: Spiced Hot Chocolate

Step: *Add 2 pieces of chocolate to the mug*

VLM: Video-LLaVa

Prompt: Does the person add 2 pieces of chocolate to the mug while performing Spiced Hot Chocolate recipe?

VLM: TimeChat

Prompt: You are given a cooking video. Please watch the video and answer the following question: Does the person add 2 pieces of chocolate to the mug? Return the answer in the format of Yes or No.

## Error Category: Measurement Error

**VLM:** Video-LLaVa

**Template:** ASSISTANT: To {complete, cook, . . . } the recipe `recipe` , the person should {pre-pare, do, . . . } the step `recipe step` . {Does, Did, . . . } the person {measure, weigh} the {ingredient} accurately?

**VLM:** TimeChat

**Template:** You are given a cooking video. Please watch the video and answer the following question: To {complete, cook, . . . } the recipe `recipe` , the person should {prepare, do, . . . } the step `recipe step` . {Does, Did, . . . } the person {measure, weigh} the {ingredient} accurately? Return the answer in the format of Yes or No.

**Question Prompts**

**Recipe:** Cucumber Raita

**Step:** *Add 1/4 teaspoon of salt to the bowl.*

**VLM:** Video-LLaVa

**Prompt:** To make the recipe Cucumber Raita, the person should add 1/4 teaspoon of salt to the bowl. Does the person measure 1/4 teaspoon of salt accurately?

**VLM:** TimeChat

**Prompt:** You are given a cooking video. Please watch the video and answer the following question: To make the recipe Cucumber Raita, the person should add 1/4 teaspoon of salt to the bowl. Does the person measure 1/4 teaspoon of salt accurately? Return the answer in the format of Yes or No.

**Recipe:** Spiced Hot Chocolate

**Step:** *Add 1/5 teaspoon of cinnamon to the mug.*

**VLM:** Video-LLaVa

**Prompt:** To cook the recipe Spiced Hot Chocolate, the person should add 1/5 teaspoon of cinnamon to the mug. Does the person measure 1/5 teaspoon of cinnamon correctly?

**VLM:** TimeChat

**Prompt:** You are given a cooking video. Please watch the video and answer the following question: To cook the recipe Spiced Hot Chocolate, the person should add 1/5 teaspoon of cinnamon to the mug. Does the person measure 1/5 teaspoon of cinnamon correctly? Return the answer in the format of Yes or No.

## Error Category: Temperature Error

| | |
|---|---|
| VLM: | Video-LLaVa |
| Template: | ASSISTANT: {While, When ... } the person is {performing, executing, ... } the step `recipe step` from the recipe `recipe` . Is any heating involved? if yes, then did the person adhere to the {low, medium, high} {heating, power level} settings of {microwave, stove}. |
| VLM: | TimeChat |
| Template: | You are given a cooking video. Please watch the video and answer the following question: {While, When ... } the person is {performing, executing, ... } the step `recipe step` from the recipe `recipe` . Is any heating involved? if yes, then did the person adhere to the {low, medium, high} {heating, power level} settings of {microwave, stove}. Return the answer in the format of Yes or No. |

## Question Prompts

| | |
|---|---|
| Recipe: | Cucumber Raita |
| Step: | *Peel a cucumber* |
| VLM: | Video-LLaVa |
| Prompt: | While the person is peeling a cucumber for making Cucumber Raita. Is any heating involved? If yes, did the person adhere to the heating settings? |
| VLM: | TimeChat |
| Prompt: | You are given a cooking video. Please watch the video and answer the following question: While the person is peeling a cucumber for making Cucumber Raita. Is any heating involved? If yes, did the person adhere to the heating settings? Return the answer in the format of Yes or No. |

| | |
|---|---|
| Recipe: | Spiced Hot Chocolate |
| Step: | *Heat the contents of the mug for 1 minute and serve.* |
| VLM: | Video-LLaVa |
| Prompt: | When the person has to heat the contents of the mug for 1 minute and serve for cooking Spiced Hot Chocolate, is any heat required? If yes, did the person adhere to the high heat setting of microwave? |
| VLM: | TimeChat |
| Prompt: | You are given a cooking video. Please watch the video and answer the following question: When the person has to heat the contents of the mug for 1 minute and serve for cooking Spiced Hot Chocolate, is any heat required? If yes, did the person adhere to the high heat setting of microwave? Return the answer in the format of Yes or No. |

## Error Category: Timing Error

**VLM:** Video-LLaVa

**Template:** To {cook, make, ... } `recipe` , the person {should, has ... } to `recipe step` . {Should, Does} the person `recipe step` {perform, make, ... } for a {specific, certain} time?

**VLM:** TimeChat

**Template:** You are given a cooking video. Please watch the video and answer the following question: To {cook, make, ... } `recipe` , the person {should, has ... } to `recipe step` . {Should, Does} the person `recipe step` {perform, make, ... } for a {specific, certain} time? Return the answer in the format of Yes or No.

**Question Prompts**

**Recipe:** Butter Corn Cup

**Step:** *Microwave the corn for 2 minutes.*

**VLM:** Video-LLaVa

**Prompt:** To make Butter Corn Cup, the person should microwave the corn for 2 minutes. Did the person microwave the corn for 2 minutes?

**VLM:** TimeChat

**Prompt:** You are given a cooking video. Please watch the video and answer the following question: To make Butter Corn Cup, the person should microwave the corn for 2 minutes. Did the person microwave the corn for 2 minutes? Return the answer in the format of Yes or No.

**Recipe:** Spiced Hot Chocolate

**Step:** *Heat the contents of the mug for 1 minute and serve.*

**VLM:** Video-LLaVa

**Prompt:** To cook Spiced Hot Chocolate, the person should heat the contents of the mug for 1 minute and serve. Does the person heat the contents of the mug for 1 minute before serving?

**VLM:** TimeChat

**Prompt:** You are given a cooking video. Please watch the video and answer the following question: To cook Spiced Hot Chocolate, the person should heat the contents of the mug for 1 minute and serve. Does the person heat the contents of the mug for 1 minute before serving? Return the answer in the format of Yes or No.

## B.2  Anomaly Detection

We used anomaly detection methods to classify each frame in each video as either normal or abnormal, where the latter is defined as an instance that deviates from the expected behaviour (the frame where participants made errors). Specifically, we used two self-supervised anomaly detection methods from the literature, self-supervised masked convolutional transformer block (SSMCTB) [56] and self-supervised predictive convolutional attentive block (SSPCAB) [77], and trained them on top of ResNet-50 [38], where the latter serves as a neural, image-based feature extractor. Both models were trained using reconstruction loss [56]. We used normal recordings for training and both normal and error recordings for testing. We evaluated the benchmark models using the frame-level area under the curve (AUC) and Equal Error Rate (EER) scores. Table 9 shows the results. We observe that SSMCTB is slightly better than SSPCAB. The AUC scores displayed in this context demonstrate only marginal improvement over random chance. This emphasizes the difficulty of the task and underscores the necessity for specialized approaches to recognize errors in a self-supervised manner.

Table 9: **Anomaly Detection**

| Method | AUC(%) | EER (%) |
|---|---|---|
| **SSMCTB** [56] | 50.65 | 49.65 |
| **SSPCAB** [77] | 50.25 | 49.74 |

## B.3 Supervised Error Recognition

**Task.**  We set up the error recognition task (namely, given a video segment, classify it either as error or normal) as a supervised binary classification problem. The presence of a variety of errors (that are both cascading and non-cascading in nature across the duration of the recording) makes solving this task particularly challenging. We use error annotations and mark a segment as *normal* if the corresponding step was performed correctly; else, we mark it as an *error*.

**Features.**  We obtained features from pre-trained video recognition models, namely (1) Slowfast, (2) X3D, (3) Omnivore, (4) 3D Resnet, and (5) Imagebind. Since the feature extractors require fixed-sized inputs (they are neural networks), we divided each video segment into contiguous 1-second sub-segments. The video segment may not always be perfectly divisible by 1 second as the last sub-segment might be shorter than 1 second. To make it uniform, we used zero padding; namely, we added zeros at the end of the sub-segment and extended its duration to 1 second to extract its features.

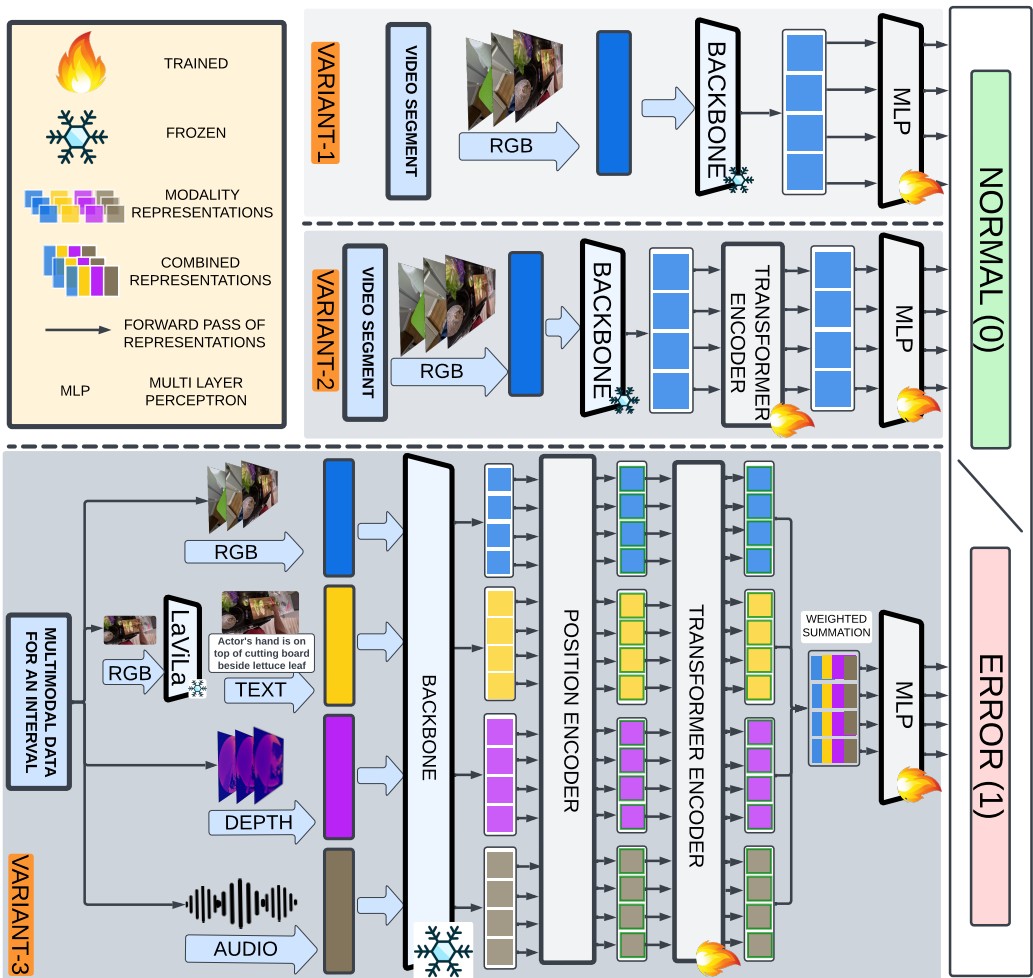

Figure 12: **SupervisedER** architectures of 3 baselines.

**Models.**  We proposed three architectural variants as baselines for Supervised Error Recognition(Fig. 12). During training, we assigned a segment's (recipe step) class label to all its 1-second sub-segments (for which features are extracted). Thus yielding the proposed splits' train, validation, and test subsets of data, which are used to learn our proposed variants of the baselines. During inference, we again divided each video segment into 1-second sub-segments and, after applying any necessary zero-padding, designated the class of the segment as the majority class of its sub-segments.

Variant-1: Below are the rough steps we followed to train our $\mathcal{V}_1$ supervised error recognition models.

- **Dataset:** We used two proposed splits, namely, Step ($\mathcal{S}$) and Recordings ($\mathcal{R}$) for training.
- **Model:** We trained our models using a Multi-Layer Perceptron (MLP) Head that includes one hidden layer. The size of this layer depends on the feature dimensions from the pre-trained video recognition models. The hidden layer is followed by a single sigmoid node.
- **Training:** We trained these models on the training subset and fine-tuned the hyperparameters using the validation subsets of the proposed splits. We maintained a uniform minibatch size of 512 instances. We employed ReLU activation functions in the hidden layers and trained using the PyTorch [67] on a single NVIDIA A40 GPU. We employed the Adam optimizer [50] for training and a learning rate of 0.001. Due to the inherent class imbalance of the constructed dataset, we used standard Binary Cross Entropy Loss (BCE Loss) with a weight of 1.5 for the positive classes. We trained all our models for 50 epochs.
- **Evaluation:** We selected the model that performed best on the validation set, evaluated it on the test set, and presented the evaluation results in the main paper.

Variant-2: Below are the rough steps we followed to train our $\mathcal{V}_2$ supervised error recognition models.

- **Dataset:** We used two proposed splits, namely, Step ($\mathcal{S}$) and Recordings ($\mathcal{R}$) for training.
- **Model:** We leveraged the strengths of transformers that facilitate using variable length inputs and allow representing time via positional encodings to train baseline error recognition models. Specifically, instead of generating predictions for 1-second sub-segments independently, we pass the sub-segment features through a transformer encoder to learn representations that are contextually aware of the entire video segment of the recipe step. Finally, we pass these representations of 1-second sub-segments through a Multi-Layer Perceptron (MLP) head that includes one hidden layer (whose size is determined by the feature dimensions of the pre-trained video recognition models) followed by a single sigmoid node.
- **Training:** We trained these models on the training subset and fine-tuned the hyperparameters using the validation subsets of the proposed splits. We maintained a uniform minibatch size of 1 video segment corresponding to a step. We trained using the PyTorch [67] on a single NVIDIA A40 GPU. We employed the Adam optimizer [50] for training and a learning rate of 1e-5. To address the inherent class imbalance of the dataset, we used standard BCE Loss with a weight of 1.5 for the positive classes and trained all our models for 50 epochs.
- **Evaluation:** We selected the model that performed best on the validation set, evaluated it on the test set, and presented the evaluation results in the main paper.

Variant-3: Below are the rough steps we followed to train our $\mathcal{V}_3$ supervised error recognition models.

- **Dataset:** We used two proposed splits, namely, Step ($\mathcal{S}$) and Recordings ($\mathcal{R}$) for training.
- **Model:** We leveraged the strengths of transformers that facilitate using variable length inputs from multiple modalities of data and allow representing time via positional encodings to train baseline error recognition models. Specifically, instead of generating predictions for a single RGB modality data, we pass the sub-segment features corresponding to RGB, audio, text and depth modalities through a transformer encoder to learn unified representations that are contextually aware of the entire video segment of the recipe step. Finally, we pass these unified representations of multiple modalities that correspond to sub-segments of the data through an MLP head that includes one hidden layer (whose size is determined by the features of the pre-trained video recognition models) followed by a single sigmoid node.
- **Training:** We trained these models on the training subset and fine-tuned the hyperparameters using the validation subsets of the proposed splits. We maintained a uniform minibatch size of 1 video segment corresponding to a step. We trained using the PyTorch [67] on a single NVIDIA A40 GPU. We employed the Adam optimizer [50] for training and a learning rate of 5e-5. To address the inherent class imbalance of the dataset, we used standard BCE Loss with a weight of 1.5 for the positive classes and trained all our models for 50 epochs.
- **Evaluation:** We selected the model that performed best on the validation set, evaluated it on the test set, and presented the evaluation results in the main paper.

Table 10: **SupervisedER** evaluation results of baselines. Variant type ($\mathcal{V}_\#$), Modality of features (M), Video (V), Audio (A), Depth (D), and Text (T).

| Split | Backbone | $\mathcal{V}_\#$ | Modality | Acc | P | R | F1 | AUC |
|-------|----------|------------------|----------|-----|---|---|----|----|
| $\mathcal{S}$ | Omnivore | $\mathcal{V}_1$ | V | 71.09 | 66.07 | 14.86 | 24.26 | 75.7 |
| | | $\mathcal{V}_2$ | V | 69.96 | 51.56 | 59.84 | 55.39 | 75.7 |
| | Slowfast | $\mathcal{V}_1$ | V | 33.54 | 31.88 | 90.6 | 47.16 | 63.06 |
| | | $\mathcal{V}_2$ | V | 68.09 | 47.69 | 24.9 | 32.72 | 67.18 |
| | X3D | $\mathcal{V}_1$ | V | 68.34 | 48 | 19.28 | 27.51 | 60.19 |
| | | $\mathcal{V}_2$ | V | 67.83 | 42.86 | 9.64 | 15.74 | 61.5 |
| | 3DResnet | $\mathcal{V}_1$ | V | 63.45 | 42.9 | 52.21 | 47.1 | 66.16 |
| | | $\mathcal{V}_2$ | V | 61.58 | 41.47 | 56.63 | 47.88 | 64.5 |
| | ImageBind | $\mathcal{V}_3$ | V | 62.78 | 38.46 | 32.13 | 35.01 | 57.03 |
| | | | A | 43.86 | 32.26 | 72.69 | 44.69 | 52.23 |
| | | | V, A | 63.28 | 40.76 | 38.96 | 39.84 | 53.87 |
| | | | V, A, T | 68.79 | 42.76 | 41.96 | 42.36 | 61.1 |
| | | | V, A, D, T | 69.4 | 50.75 | 48.96 | 49.84 | 70.41 |
| $\mathcal{R}$ | Omnivore | $\mathcal{V}_1$ | V | 59.76 | 45.31 | 58.09 | 50.91 | 63.03 |
| | | $\mathcal{V}_2$ | V | 62.3 | 46.55 | 33.61 | 39.04 | 62.27 |
| | Slowfast | $\mathcal{V}_1$ | V | 60.06 | 40.82 | 24.9 | 30.93 | 56.89 |
| | | $\mathcal{V}_2$ | V | 57.82 | 41.67 | 43.57 | 42.6 | 59.83 |
| | X3D | $\mathcal{V}_1$ | V | 54.69 | 39.6 | 49.79 | 44.12 | 54.66 |
| | | $\mathcal{V}_2$ | V | 54.25 | 40.78 | 60.58 | 48.75 | 56.58 |
| | 3DResnet | $\mathcal{V}_1$ | V | 41.88 | 37.65 | 94.19 | 53.79 | 62.42 |
| | | $\mathcal{V}_2$ | V | 57.82 | 43.56 | 58.92 | 50.09 | 59.22 |
| | ImageBind | $\mathcal{V}_3$ | V | 37.56 | 36.14 | 96.27 | 52.55 | 54.1 |
| | | | A | 39.05 | 35.67 | 89.72 | 50.54 | 54.8 |
| | | | V, A | 54.25 | 40.98 | 62.24 | 49.42 | 55.25 |
| | | | V, A, T | 64.08 | 44.15 | 58.01 | 50.13 | 58.35 |
| | | | V, A, D, T | 63.87 | 49.57 | 64.4 | 56.02 | 65.25 |

## B.4 Supervised Early Error Recognition

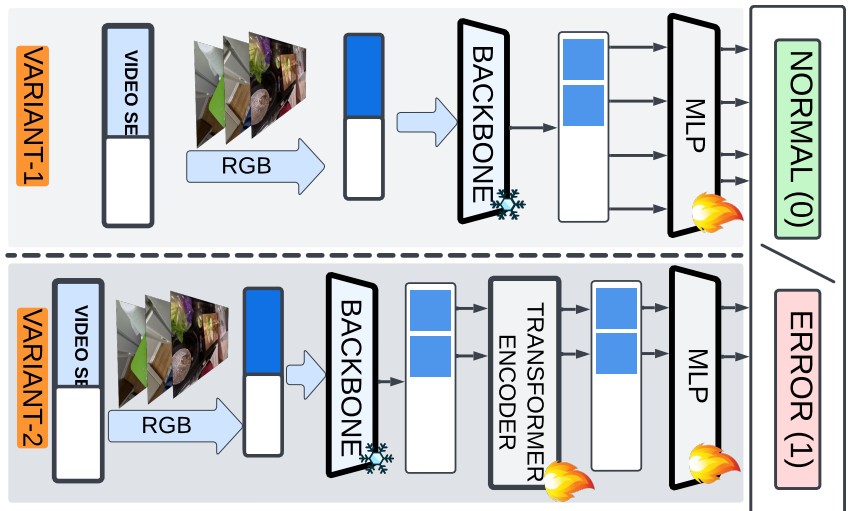

Figure 13: **Supervised Early Error Recognition** architectures of baselines.

**Task.** We set up the early error recognition task (namely, given only the first half of the video segment corresponding to a step, classify it either as error or normal) as a supervised binary classification problem. Since the model only processes the first half of the video segment, which mainly showcases the pre-conditions for an action, error recognition in this context involves ***anticipating potential errors*** that may arise from deviations in the pre-conditions of the action. Thus, early error recognition is an extremely hard setting and the presence of a variety of errors (that are both cascading and non-cascading in nature across the duration of the recording) makes it more challenging.

**Features.** We obtained features from pre-trained video recognition models, namely (1) Slowfast, (2) X3D, (3) Omnivore, (4) 3D Resnet. Since the feature extractors require fixed-sized inputs (they are neural networks), we divided each video segment into contiguous 1-second sub-segments. The video segment may not always be perfectly divisible by 1 second as the last sub-segment might be shorter than 1 second. To make it uniform, we used zero padding; namely, we added zeros at the end of the sub-segment and extended its duration to 1 second to extract its features.

**Models.** We proposed two architectural variants as baselines for Supervised Early Error Recognition(Fig. 13). During training, we assigned a segment's (recipe step) class label to all its observed 1-second sub-segments (first half of the video segments). Thus yielding the proposed splits' train, validation, and test subsets of data, which are used to train our baselines. During inference, we again divided each partially observed video segment into 1-second sub-segments and assigned the class label of the partially observed video segment as the majority class of observed sub-segments.

Variant-1: Rough steps we followed to train our $\mathcal{V}_1$ supervised early error recognition models.

- **Dataset:** We used two proposed splits, namely, Step ($\mathcal{S}$) and Recordings ($\mathcal{R}$) for training.
- **Model:** We trained our models using a Multi-Layer Perceptron (MLP) Head that includes one hidden layer. The size of this layer depends on the feature dimensions from the pre-trained video recognition models. The hidden layer is followed by a single sigmoid node.
- **Training:** We trained these models on the training subset and fine-tuned the hyperparameters using the validation subsets of the proposed splits. We maintained a uniform minibatch size of 512 instances. We employed ReLU activation functions in the hidden layers and trained using the PyTorch [67] on a single NVIDIA A40 GPU. We employed the Adam optimizer [50] for training and a learning rate of 1e-4. Due to the inherent class imbalance of the constructed dataset, we used standard Binary Cross Entropy Loss (BCE Loss) with a weight of 1.5 for the positive classes. We trained all our models for 50 epochs.

– **Evaluation:** We selected the model that performed best on the validation set, evaluated it on the test set, and presented the evaluation results in Table 11.

Variant-2: Rough steps we followed to train our $\mathcal{V}_2$ supervised error recognition models.

– **Dataset:** We used two proposed splits, namely, Step ($\mathcal{S}$) and Recordings ($\mathcal{R}$) for training.

– **Model:** Instead of generating predictions for 1-second sub-segments independently, we pass the sub-segment features through a transformer encoder to learn representations that are contextually aware of the entire video segment of the recipe step. Finally, we pass these representations of 1-second sub-segments through a Multi-Layer Perceptron (MLP) head that includes one hidden layer (whose size is determined by the feature dimensions of the pre-trained video recognition models) followed by a single sigmoid node.

– **Training:** We trained these models on the training subset and fine-tuned the hyperparameters using the validation subsets of the proposed splits. We maintained a uniform minibatch size of 1 video segment corresponding to a step. We trained using the PyTorch [67] on a single NVIDIA A40 GPU. We employed the Adam optimizer [50] for training and a learning rate of 1e-5. To address the inherent class imbalance of the dataset, we used standard BCE Loss with a weight of 1.5 for the positive classes and trained all our models for 50 epochs.

– **Evaluation:** We selected the model that performed best on the validation set, evaluated it on the test set, and presented the evaluation results in Table 11.

Table 11: **SupervisedEER** evaluation results of baselines. Variant type ($\mathcal{V}_\#$), Modality of features (M), Video (V), Audio (A), Depth (D), and Text (T).

| Split | Backbone | $\mathcal{V}_\#$ | Modality | Acc | P | R | F1 | AUC |
|---|---|---|---|---|---|---|---|---|
| $\mathcal{S}$ | Omnivore | $\mathcal{V}_1$ | V | 69.59 | 75 | 3.61 | 6.9 | 72.9 |
| | | $\mathcal{V}_2$ | V | 69.21 | 50 | 1.2 | 2.34 | 73.48 |
| | Slowfast | $\mathcal{V}_1$ | V | 67.96 | 47.15 | 23.29 | 31.18 | 67.23 |
| | | $\mathcal{V}_2$ | V | 69.09 | 50 | 1.6 | 3.1 | 62.72 |
| | X3D | $\mathcal{V}_1$ | V | 65.96 | 41.73 | 23.29 | 29.9 | 59.27 |
| | | $\mathcal{V}_2$ | V | 68.71 | 45.45 | 2.01 | 3.85 | 56.5 |
| | 3DResnet | $\mathcal{V}_1$ | V | 68.46 | 47.89 | 13.65 | 21.25 | 63.79 |
| | | $\mathcal{V}_2$ | V | 68.84 | 50 | 0.4 | 0.8 | 60.76 |
| $\mathcal{R}$ | Omnivore | $\mathcal{V}_1$ | V | 64.08 | 50 | 2.07 | 3.98 | 64.73 |
| | | $\mathcal{V}_2$ | V | 64.08 | 50 | 0.41 | 0.82 | 65.43 |
| | Slowfast | $\mathcal{V}_1$ | V | 63.79 | 33.33 | 0.83 | 1.62 | 53.11 |
| | | $\mathcal{V}_2$ | V | 63.93 | 42.86 | 1.24 | 2.42 | 53.38 |
| | X3D | $\mathcal{V}_1$ | V | 63.34 | 46.27 | 12.86 | 20.13 | 55.24 |
| | | $\mathcal{V}_2$ | V | 63.64 | 28.57 | 0.83 | 1.61 | 56.84 |
| | 3DResnet | $\mathcal{V}_1$ | V | 63.19 | 43.75 | 8.71 | 14.53 | 53.47 |
| | | $\mathcal{V}_2$ | V | 63.04 | 37.04 | 4.15 | 7.46 | 54.73 |

**Remark:** We observed that there is a significant drop in performance metrics of all our models compared to the Supervised Error Recognition task. We also note that our $\mathcal{V}_2$ models exhibited lower performance than our $\mathcal{V}_1$ models. We attribute this drop to the extremely noisy signal (due to the observation of only the pre-conditions of actions) used to recognize errors in the recordings.

## B.5 Multi-Step Localization

Multi-Step localization (MSL) entails both recognizing and localization of steps within a procedural activity. For this task, we leverage features extracted from pre-trained video recognition models and train an ActionFormer head to manage the processes of step recognition and localization. In the main text, we detailed the experimental evaluations of MSL and Robust MSL. Additionally, we trained models using features extracted with Omnivore as the backbone for video segments of 1-second, 3-second, and 4-second lengths, and we present the results in Table 12. We observed a performance enhancement in the model as the length of the video segments used for feature extraction increased.

Table 12: **MSL** using Omnivore features corresponding to video segments of varying lengths

| $\mathcal{B}$ | $\mathcal{D}$ | $\mathcal{I}_t = 0.1$ | | | $\mathcal{I}_t = 0.3$ | | | $\mathcal{I}_t = 0.5$ | | |
|---|---|---|---|---|---|---|---|---|---|---|
| | | mAP | R@1 | R@5 | mAP | R@1 | R@5 | mAP | R@1 | R@5 |
| | $\mathcal{E}$ | 67.51 | 64.45 | 62.31 | 85.32 | 82.82 | 78.11 | 38.32 | 36.54 | 33.41 |
| $\mathcal{O}_{1s}$ | $\mathcal{P}$ | **75.96** | **73.35** | **70.34** | **92.14** | **90.51** | **88.24** | **45.82** | **44.12** | **41.16** |
| | $\mathcal{R}$ | 73.71 | 71.45 | 68.14 | 92.08 | 89.82 | 86.38 | 42.76 | 40.52 | 37.19 |
| | $\mathcal{E}$ | 72.99 | 70.05 | 66.57 | 86.03 | 83.68 | 81.02 | 43.47 | 41.83 | 38.87 |
| $\mathcal{O}_{3s}$ | $\mathcal{P}$ | **78.63** | **76.96** | **74.61** | **93.27** | **91.23** | **89.18** | **50.25** | **48.54** | **44.85** |
| | $\mathcal{R}$ | 76.82 | 74.90 | 71.94 | 91.33 | 89.61 | 88.11 | 49.23 | 47.84 | 44.76 |
| | $\mathcal{E}$ | 71.85 | 69.79 | 64.93 | 88.12 | 86.33 | 83.15 | 43.13 | 41.54 | 38.95 |
| $\mathcal{O}_{4s}$ | $\mathcal{P}$ | **79.33** | **77.39** | **74.24** | **93.46** | **91.67** | **89.95** | **50.69** | **49.49** | **46.19** |
| | $\mathcal{R}$ | 78.61 | 76.59 | 73.81 | 93.04 | 90.99 | 88.80 | 50.24 | 48.62 | 45.64 |

Table 13: **MSL** evaluation results.

| $\mathcal{B}$ | $\mathcal{D}$ | $\mathcal{I}_t = 0.1$ | | | $\mathcal{I}_t = 0.3$ | | | $\mathcal{I}_t = 0.5$ | | |
|---|---|---|---|---|---|---|---|---|---|---|
| | | mAP | R@1 | R@5 | mAP | R@1 | R@5 | mAP | R@1 | R@5 |
| | $\mathcal{E}$ | 25.98 | 54.82 | 77.59 | 23.75 | 48.38 | 72.19 | 19.59 | 38.44 | 61.87 |
| **3D Resnet** | $\mathcal{P}$ | 29.29 | 63.07 | 88.96 | 27.71 | 56.60 | 84.75 | 23.21 | 46.79 | 76.86 |
| | $\mathcal{R}$ | 29.39 | 61.14 | 85.41 | 27.89 | 55.82 | 82.17 | 23.97 | 46.54 | 73.29 |
| | $\mathcal{E}$ | 27.68 | 55.73 | 77.45 | 25.51 | 48.98 | 70.90 | 21.09 | 37.82 | 60.58 |
| **Slowfast** | $\mathcal{P}$ | 32.77 | 63.22 | 90.43 | 31.21 | 58.82 | 86.82 | 27.25 | 50.70 | 79.49 |
| | $\mathcal{R}$ | 32.90 | 63.97 | 89.29 | 31.47 | 59.26 | 85.32 | 27.89 | 51.62 | 77.27 |
| | $\mathcal{E}$ | 28.12 | 51.76 | 73.00 | 26.38 | 46.16 | 67.87 | 21.35 | 37.12 | 57.81 |
| **VideoMAE** | $\mathcal{P}$ | 38.86 | 64.86 | 84.05 | 37.41 | 60.32 | 80.63 | 32.24 | 51.46 | 71.88 |
| | $\mathcal{R}$ | 37.44 | 63.08 | 80.90 | 35.11 | 57.30 | 77.38 | 30.76 | 49.19 | 69.43 |
| | $\mathcal{E}$ | 40.40 | 67.51 | 87.69 | 38.32 | 62.31 | 82.82 | 33.41 | 53.01 | 72.85 |
| **Omnivore** | $\mathcal{P}$ | 48.16 | 75.96 | 93.41 | 45.82 | 70.34 | 90.51 | 41.16 | 62.00 | 84.73 |
| | $\mathcal{R}$ | 44.81 | 73.71 | 93.34 | 42.76 | 68.14 | 89.82 | 37.19 | 56.93 | 81.86 |

Table 14: **RobustMSL** evaluation results.

| $\mathcal{B}$ | $\mathcal{D}$ | $\mathcal{T}$ | $\mathcal{I}_t = 0.1$ | | | $\mathcal{I}_t = 0.3$ | | | $\mathcal{I}_t = 0.5$ | | |
|---|---|---|---|---|---|---|---|---|---|---|---|
| | | | mAP | R@1 | R@5 | mAP | R@1 | R@5 | mAP | R@1 | R@5 |
| **VideoMAE** | $\mathcal{E}$ | $\mathcal{T}_n$ | 24.44 | 38.22 | 52.48 | 22.97 | 34.77 | 49.51 | 18.67 | 28.57 | 42.68 |
| | | $\mathcal{T}_e$ | 7.53 | 13.54 | 20.52 | 6.93 | 11.4 | 18.36 | 5.63 | 8.55 | 15.13 |
| | $\mathcal{P}$ | $\mathcal{T}_n$ | 26.78 | 37.43 | 46.28 | 25.68 | 34.79 | 44.6 | 22.02 | 29.43 | 39.81 |
| | | $\mathcal{T}_e$ | 16.98 | 27.43 | 37.76 | 16.46 | 25.53 | 36.03 | 14.64 | 22.03 | 32.07 |
| | $\mathcal{R}$ | $\mathcal{T}_n$ | 26.27 | 37.15 | 46.93 | 24.71 | 34.06 | 45.03 | 21.51 | 29.36 | 40.44 |
| | | $\mathcal{T}_e$ | 15.43 | 25.94 | 33.97 | 14.44 | 23.23 | 32.35 | 12.96 | 19.83 | 28.99 |
| **Omnivore** | $\mathcal{E}$ | $\mathcal{T}_n$ | 34.65 | 47.91 | 60.63 | 33.06 | 44.77 | 58.36 | 28.59 | 38.38 | 51.9 |
| | | $\mathcal{T}_e$ | 12.51 | 19.6 | 27.06 | 11.66 | 17.54 | 24.45 | 9.94 | 14.63 | 20.96 |
| | $\mathcal{P}$ | $\mathcal{T}_n$ | 32.5 | 44.45 | 52.47 | 31.13 | 41.53 | 50.91 | 28.39 | 37.03 | 47.97 |
| | | $\mathcal{T}_e$ | 21.28 | 31.51 | 40.93 | 20.12 | 28.81 | 39.6 | 18.08 | 24.96 | 36.77 |
| | $\mathcal{R}$ | $\mathcal{T}_n$ | 30.22 | 42.43 | 52.11 | 28.94 | 39.47 | 50.49 | 25.15 | 32.65 | 46.51 |
| | | $\mathcal{T}_e$ | 19.54 | 31.28 | 41.24 | 18.4 | 28.66 | 39.33 | 16.27 | 24.28 | 35.35 |

**Extended Analysis.** In Tables 13 & 14, we note that when data is split by environments, with the test set comprising new environments, the models, struggle to recognize the steps performed in the videos. As we increase thresholded Intersection Over Union ($\mathcal{I}_t$), we observe a drop in the performance of the models, thus signifying the low confidence in the prediction of the current steps.

## B.6 Procedure Learning

Given long, untrimmed videos of procedural activities where the sequences of steps can be performed in multiple orders, self-supervised procedure learning entails the identification of relevant frames across videos of activity and the estimation of sequential steps required to complete the activity. Thus, the task entails the identification of key steps and their sequence to complete an activity. To benchmark procedure learning, we used normal recordings from our dataset and assessed the performance of recently proposed methods [3, 16]. In the main text, we presented results when evaluated on only 5 recipes. Here, in Table 15, we present the evaluation results on all 24 recipes. The results in Table 15 showcase the performance of models trained using methods $\mathcal{M}_1$ [16] and $\mathcal{M}_2$ [3]. Where $\mathcal{M}_1$ employs Cycleback Regression Loss ($\mathcal{C}$) and $\mathcal{M}_1$ employs a combination of both Cycleback Regression Loss ($\mathcal{C}$) and Contrastive - Inverse Difference Moment Loss ($\mathscr{C}$). It is important to note that we only train embedder networks using these loss functions and maintain the Pro-Cut Module (PCM) for assigning frames to key steps. In Table 15, $\mathcal{P}$ represents precision, $R$ represents recall, and $I$ represents IOU.

Table 15: **Self-Supervised Procedure Learning** evaluation results on the selected 24 recipes.

| Recipe | Random | | | $\mathcal{M}_1$ | | | $\mathcal{M}_2$ | | |
|---|---|---|---|---|---|---|---|---|---|
| | $\mathcal{P}$ | $\mathcal{R}$ | $\mathcal{I}$ | $\mathcal{P}$ | $\mathcal{R}$ | $\mathcal{I}$ | $\mathcal{P}$ | $\mathcal{R}$ | $\mathcal{I}$ |
| BlenderBananaPancakes | 7.40 | 3.83 | 2.26 | 12.65 | 9.50 | 5.16 | 15.54 | 9.96 | 5.72 |
| BreakfastBurritos | 9.66 | 4.04 | 2.59 | 18.72 | 11.46 | 6.77 | 16.58 | 10.77 | 5.87 |
| BroccoliStirFry | 4.21 | 3.81 | 1.73 | 9.92 | 9.11 | 3.93 | 8.20 | 8.10 | 3.85 |
| ButterCornCup | 8.37 | 3.91 | 2.16 | 13.82 | 11.85 | 5.79 | 15.07 | 12.30 | 5.82 |
| CapreseBruschetta | 9.34 | 3.96 | 2.52 | 25.55 | 12.89 | 7.52 | 20.53 | 9.09 | 5.59 |
| CheesePimiento | 9.10 | 3.87 | 2.41 | 19.74 | 10.48 | 6.44 | 17.49 | 10.32 | 6.26 |
| Coffee | 6.54 | 3.87 | 2.17 | 13.68 | 9.91 | 5.49 | 15.76 | 10.25 | 5.63 |
| CucumberRaita | 8.90 | 3.64 | 2.44 | 13.58 | 7.92 | 5.14 | 16.15 | 9.97 | 6.09 |
| DressedUpMeatballs | 7.28 | 3.80 | 2.26 | 15.20 | 10.80 | 6.05 | 17.59 | 10.27 | 5.81 |
| HerbOmeletWithFriedTomatoes | 6.82 | 4.05 | 1.98 | 14.66 | 14.98 | 5.50 | 14.64 | 11.34 | 6.29 |
| MicrowaveEggSandwich | 8.81 | 3.98 | 2.61 | 16.25 | 10.44 | 6.16 | 19.16 | 11.29 | 6.99 |
| MicrowaveFrenchToast | 9.03 | 3.74 | 2.49 | 16.82 | 7.90 | 5.07 | 17.31 | 8.82 | 5.66 |
| MicrowaveMugPizza | 7.53 | 3.90 | 2.38 | 12.82 | 9.78 | 5.27 | 12.69 | 9.18 | 5.18 |
| MugCake | 5.45 | 4.00 | 2.12 | 16.12 | 12.95 | 6.87 | 10.32 | 8.85 | 4.40 |
| PanFriedTofu | 5.35 | 3.97 | 1.54 | 8.86 | 10.39 | 3.75 | 9.34 | 12.44 | 3.87 |
| Pinwheels | 6.54 | 4.28 | 2.13 | 13.58 | 11.96 | 5.92 | 16.08 | 13.06 | 7.05 |
| Ramen | 6.85 | 4.12 | 1.87 | 11.09 | 9.97 | 4.48 | 12.90 | 10.92 | 5.07 |
| SautedMushrooms | 6.08 | 3.81 | 2.02 | 15.06 | 12.22 | 6.16 | 19.54 | 13.83 | 7.42 |
| ScrambledEggs | 4.74 | 3.95 | 1.89 | 11.11 | 11.08 | 5.27 | 11.70 | 10.96 | 5.27 |
| SpicedHotChocolate | 14.08 | 3.82 | 3.09 | 29.82 | 10.58 | 8.49 | 29.79 | 11.04 | 8.74 |
| SpicyTunaAvocadoWraps | 6.25 | 3.90 | 2.21 | 15.62 | 10.52 | 5.67 | 12.47 | 9.61 | 5.25 |
| TomatoChutney | 5.45 | 3.89 | 1.85 | 12.25 | 10.68 | 5.42 | 12.25 | 10.68 | 5.42 |
| TomatoMozzarellaSalad | 10.88 | 3.91 | 2.38 | 19.77 | 10.21 | 6.01 | 19.20 | 10.48 | 5.96 |
| Zoodles | 7.91 | 4.08 | 2.22 | 18.32 | 12.80 | 6.37 | 18.32 | 12.80 | 6.37 |
| Average | **7.61** | **3.92** | **2.22** | **15.62** | **10.85** | **5.78** | **15.78** | **10.68** | **5.82** |

## C  Data

### C.1  Data Collection Planning

Our objective is to capture data that aids in detecting, segmenting, and analyzing errors that occur during the execution of long procedural tasks. To accomplish this, we need to address the following:

1. **What to record:** Specifically, select the domain and tasks (such as recipes).
2. **How to record:** Choice of appropriate sensors and development of data capturing system.
3. **Whom to record:** This entails participant selection and training.

#### C.1.1  What to record?

Current procedural activity datasets encompass recorded and curated ones from crowd-sourced online platforms. Amongst the recorded datasets, Breakfast [51], 50Salads [87], CMU-MMAC [13], and GTEA [21] capture people performing cooking activities, and Assembly-101 [19], EPIC-TENTS [44] and MECCANO [75] capture people performing activities related to assembly of toys, tents and lego blocks, respectively. Curated datasets like COIN [89], CrossTask [105], and HowTo100M [60] encompass a wide variety of activities from different domains. We introduced a new perspective on understanding procedural activities from the lens of errors made while performing procedural tasks. We embark on an investigation into this new idea by choosing cooking as the domain of interest. This careful choice stems from the fact that cooking activities often encompass complex procedures and provide an opportunity to capture a plethora of potential, predominantly benign errors.

Table 16: Selected recipes categorized based on the type of required heating instrument.

| Heating Instrument | Recipe | | Heating Instrument | Recipe |
|---|---|---|---|---|
| Kettle | Coffee | | Nothing | Pinwheels |
| Microwave | Breakfast Burritos | | | Spicy Tuna Avocado Wraps |
| | Butter Corn Cup | | | Tomato Mozzarella Salad |
| | Cheese Pimiento | | Pan | Blender Banana Pancakes |
| | Dressed Up Meatballs | | | Broccoli Stir Fry |
| | Microwave Egg Sandwich | | | Caprese Bruschetta |
| | Microwave French Toast | | | Herb Omelet with Fried Tomatoes |
| | Microwave Mug Pizza | | | Pan Fried Tofu |
| | Mug Cake | | | Sauteed Mushrooms |
| | Ramen | | | Scrambled Eggs |
| | Spiced Hot Chocolate | | | Tomato Chutney |
| Nothing | Cucumber Raita | | | Zoodles |

**Recipes & Task Graphs**  We have carefully selected 24 diverse recipes from WikiHow (Table 16) that represent various cuisines and require different culinary tools during preparation. Each recipe in our selected set can be subdivided into several atomic steps, where each step involves performing a specific sub-task in the recipe. In general, most recipes available on the web list these sub-tasks in a specific order. However, common sense tells us that each recipe can often be described by a partial order over the sub-tasks rather than a total order. More formally, we use a task graph to represent the partial order over the steps. Each node in the task graph corresponds to a step, and a directed edge between node $i$ and node $j$ denotes that step $i$ must be done before step $j$ (namely $i$ is a pre-condition of $j$). For our selected recipes, the corresponding task graphs are directed acyclic graphs, and therefore a topological sort over them is a valid execution of the recipe. Our task graphs also include two dummy nodes, "START" and "END", which denote the start and end of recipes, respectively and ensure that our task graphs always have one start node and one terminal node.

To simplify the complexity of a recipe, we have adopted a technique that uses a flow graph structure [96] to represent the dependencies between steps (think of it like a flowchart but designed for recipes). This approach helps us establish a precise connection between actions and their consequences. Using an action-centric graph, we emphasize the steps involved in the procedure and illustrate the sequence of operations in an easy-to-understand manner. Each action influences the subsequent ones, effectively demonstrating the interdependencies between tasks. Figure 15 presents an example of a task graph.

We illustrate the process we used to convert a recipe to a task graph using the recipe *Blender Banana Pancakes* (see figures 14 and 15 for a visual guide). Given the recipe description, we first identify all

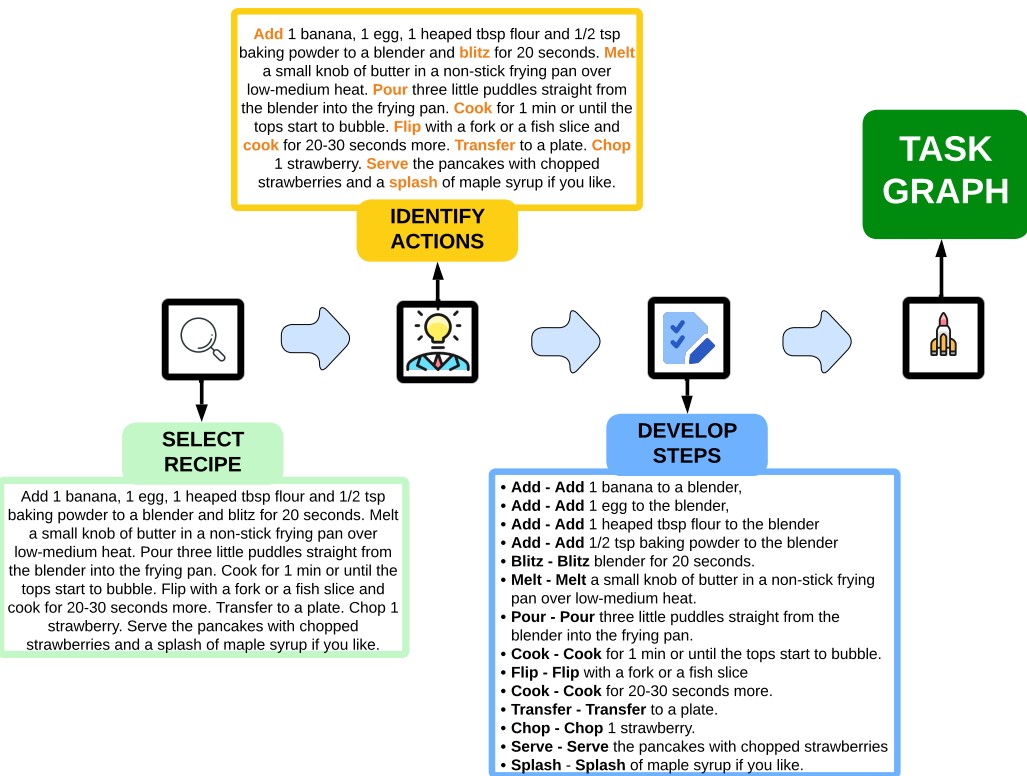

Figure 14: **TaskGraphGeneration.** This figure demonstrates the four-step process used to create an action-centric graph for a recipe, using an example. Given the recipe text as shown in *select recipe*, we identify and mark all the actions necessary for the execution of the recipe as shown in *identify actions*. Once these actions are identified, we develop them into steps (as shown in *develop steps*) ensuring each step encompasses only one of the previously identified actions. These steps are used to construct an action-centric graph for the recipe resulting in a structure as depicted in Figure 15

the actions necessary to complete the recipe and develop steps based on the identified actions, where each step contains only one among the identified actions, as shown in figure 14. After we develop steps, we use a relationship annotation tool[19] to represent the implicit order constraints amongst the developed steps. The creation of action-centric graphs serves multiple purposes. These graphs can be utilized to prepare recipe scripts with various orders while still strictly adhering to the constraints present in the graph. Moreover, given a recording, the graph can be used to verify if the individual followed the correct sequence of actions based on the inherent graph structure.ipe *Blender Banana Pancakes*, the developed steps from 14, when represented as an action-centric graph, result in Fig. 15.

In the future, we envision using our dataset to construct more fine-grained task graphs where the meaning of the steps is taken into account and how the step changes the environment (post-condition for a step). In literature, different methods have been proposed to illustrate procedural activities using task graphs and their variations, such as FlowGraphs [96], Recipe Programs [66], ConjugateTaskGraphs [36], and ActionDynamicTaskGraphs [59] and our dataset can be used to learn these task graphs in an unsupervised manner (or one can use the semantics of these various task graphs to label the videos and solve the problem in a supervised manner).

---

[19]https://www.lighttag.io/

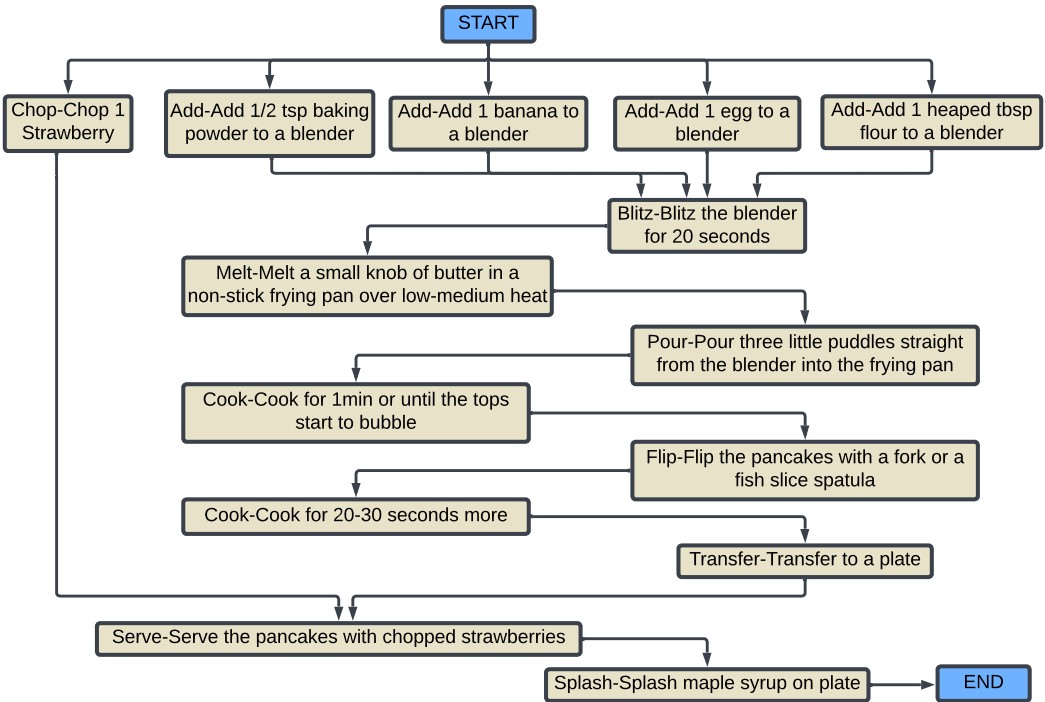

Figure 15: This graph displays the implicit dependency structure of the recipe **Blender Banana Pancakes** where the content of each node can be interpreted as {{action}-{step}} where {action} presents the description of the necessary action to be performed, and {step} presents the description as presented in the recipe text that encompasses the action, ingredients and their quantity required for the execution of the action, necessary tools used in the execution of the action, constraints on the duration of the action, how it is performed, why it is performed and other necessary settings of the environment. e.g., {Add} - {Add one banana to a blender}; here add is the necessary action and the step: Add one banana to a blender describes the action (adding), ingredient (banana), quantity (1)

### C.1.2 How to record?

**Sensors.** Recognizing the limitations of the Hololens2 augmented reality device in capturing data, despite its advanced technology, we decided to employ a dual-device strategy [20]. While the Hololens2 offers a wealth of data from various sensors, we faced two main challenges. First, the limited field of view of the RGB camera inhibits comprehensive data capture. Second, utilizing all the secondary sensors of the Hololens2 requires operating in research mode, which, unfortunately, leads to a significant frame rate reduction for other sensory data, such as depth and monochrome cameras, when we increase the quality of the captured RGB frames.

To address these issues, we integrated a GoPro into our data-capturing system. Positioned above the Hololens2 on a head mount worn by the participant, the GoPro records 4K videos at 30 frames per second, offering a wider field of view compared to that of the Hololens2's RGB frames. This setup provides us with a more detailed perspective on the participant's activities. We use the Hololens2 in research mode to obtain a diverse range of data, including depth streams and spatial information such as head and hand movements. Additionally, we collect data from three Inertial Measurement Unit (IMU) sensors: an accelerometer, a gyroscope, and a magnetometer. This combined approach enables us to capture complete, high-quality activity data.

**Data Capturing System.** We have designed a versatile and expandable system for recording procedural activities, which can be readily adapted to meet various needs. This system has two

---

[20]Although we use a dual-device strategy to record activities, it's important to note that these devices aren't synchronized prior to the start of the recording process. Instead, captured footage from both devices is programmatically synchronized during post-processing using the associated timestamps

distinct use cases: (1) as a standalone **user application** specifically designed for procedural activities and (2) as a comprehensive, plug-and-play system that functions beyond being just a user application. In its first mode, the application serves a dual role: primarily as a display interface [21] for the procedure of the activity and secondarily as a tool for noting and updating any errors made during the execution of the activity. In its second mode, the system is equipped to **capture data streams** from various sensors and allows for easy connection and configuration. This dual functionality enhances the system's adaptability, making it an efficient tool for a wide range of procedural activities.

**(1) User Application.** Using several illustrative snippets, we will briefly explain how our system can be used as a user interface to capture complex procedural activities, including errors. This process within our system is divided into four stages to facilitate data collection from participants:

Stage-1. First stage (Figure 16) presents the participant with a list of activities to the left. Upon selection, corresponding steps for the selected activity are then displayed on the right side of the page.

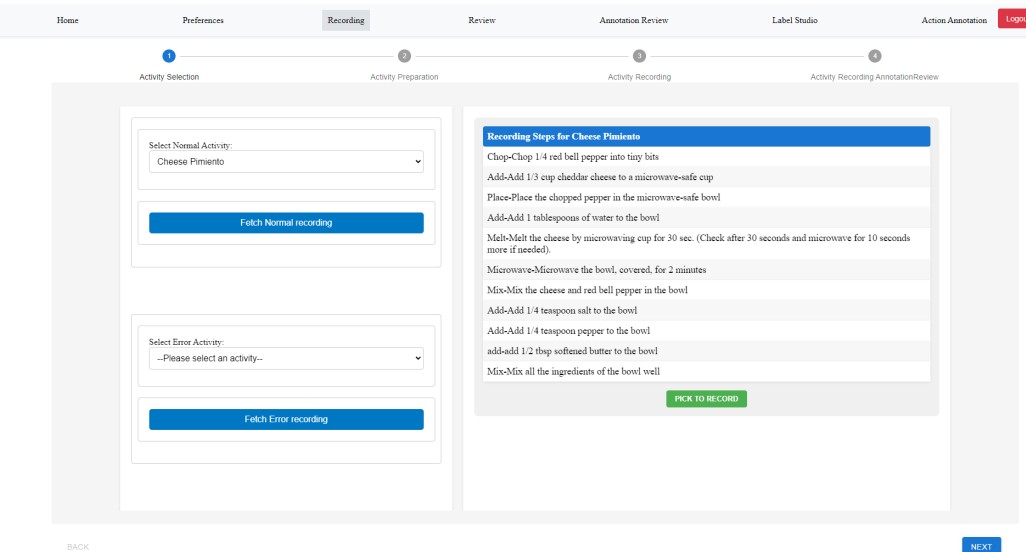

Figure 16: **Stage-1.** The participant selects the activity they wish to perform from the options presented on the left. Once a selection is made, the necessary steps for the chosen recipe will be displayed.

We provide two methods for presenting the steps of an activity, based on the input received when information about the activities is uploaded to the database:

- **Recipe Text**: If the activity's input is in plain text format, we display the text as provided.
- **Task Graph**: If the input is an action-centric task graph, we present a valid sequence of steps that conforms to the constraints defined by the graph.

Stage-2. **Activity Preparation Stage.** Although optional for a normal recording, its primary function is to prepare a script to execute during an error recording. One of our approaches to capturing error recordings involves providing participants with an interface to contemplate the errors they intend to make and modify the description of the steps for a particular activity recording session. As illustrated in Figure 17, participants can update each step based on different types of errors categorized as described above. When the participant records, they will see the updated step description as part of the sequence of steps. Moreover, GPT-4 has provided suggestions on potential errors that may occur during the activity, now available as static hint options for this recipe. However, we have observed that these *generic errors provided by GPT-4 are not particularly helpful*, as participants only considered them for script preparation in 20% of cases.

Stage-3. We present the participants with the sequence of steps (topological order of the task graph) for the selected activity that they will perform as illustrated in Figure 18.

---

[21] Please view the tablet that displays the interface, as shown in video snippets posted on our website

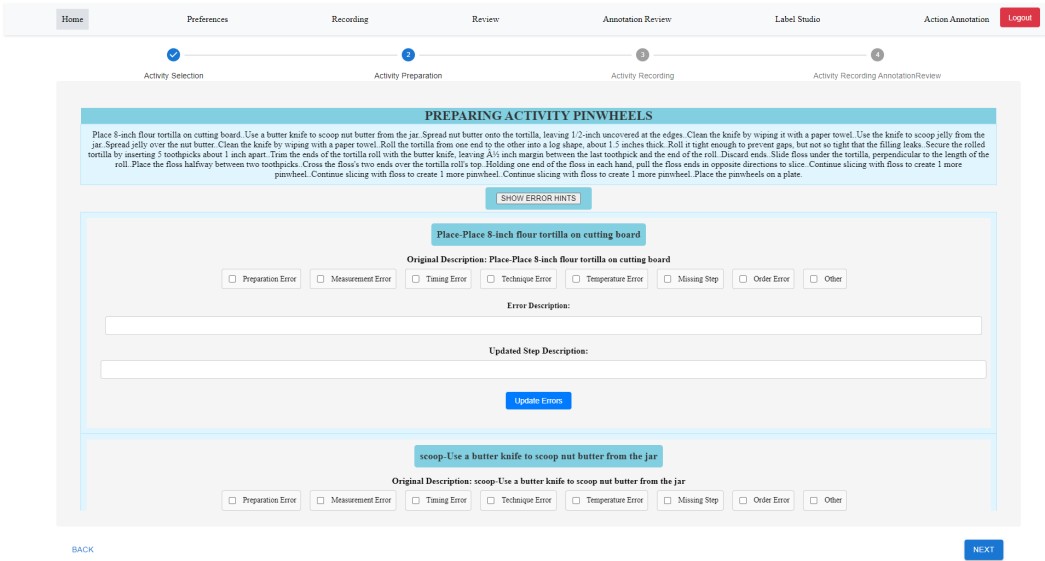

Figure 17: **Stage-2.** Interface enables participants to update step descriptions and prepare error scripts.

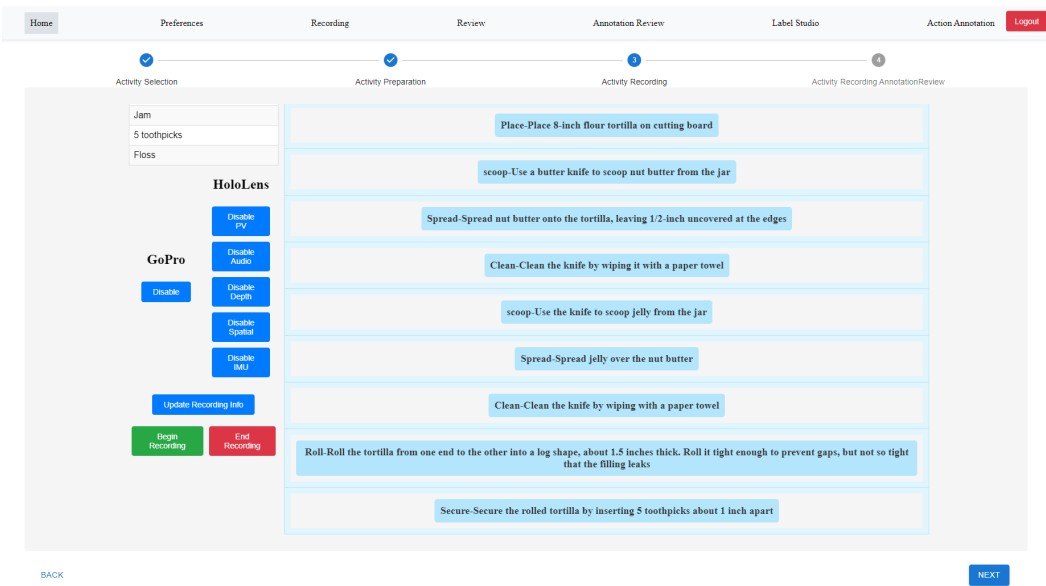

Figure 18: **Stage-3.** Displays the necessary steps to complete an activity.

Stage-4. After the data is captured either using our system or from a standalone recording system, we provide an interface to participants to review the recording they performed and correspondingly update any unplanned errors they make while performing the activity. In one of our strategies for capturing error recordings, we asked participants to induce errors *impromptu* while performing the activity. Here participants are given a series of steps corresponding to the task graph's topological order. Subsequently, participants updated information about errors they made while performing the recipe. Figure 19 presents a snippet where the participant updates one of the errors made while performing the recipe *Caprese Bruschetta*

**(2) Data Capturing Application.** The standalone application discussed earlier can be converted into a data-capturing application by integrating several plug-and-play modules. We constructed both user and data capturing applications using the software components illustrated in Figure 20.

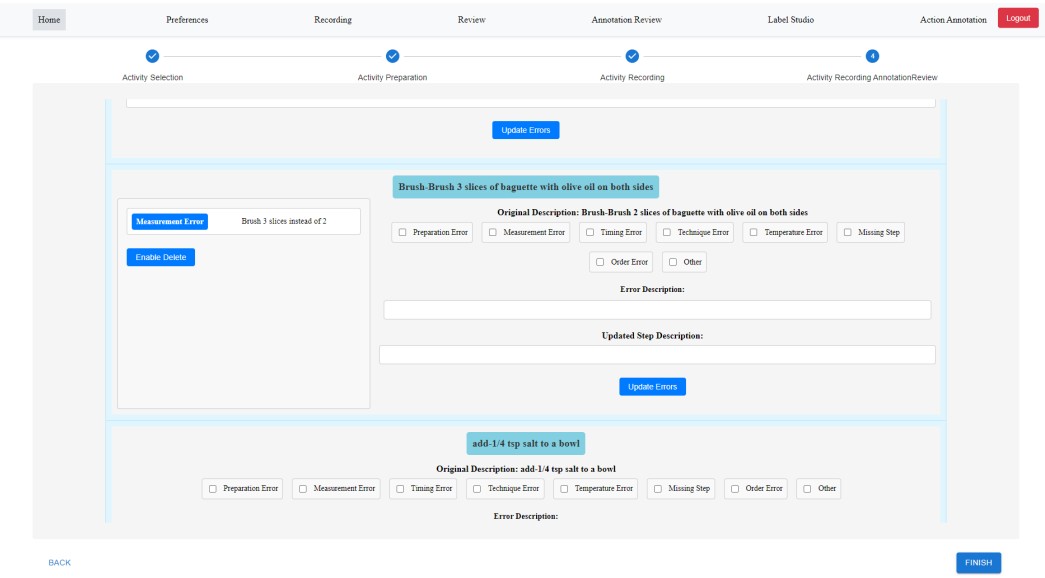

Figure 19: **Stage-4.** Similar to Stage 2, the interface allows participants to update the errors induced.

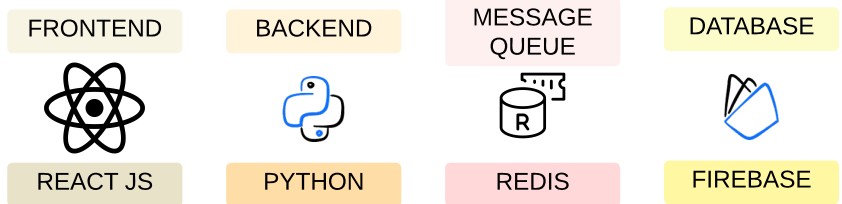

Figure 20: **Software Components** used to build the proposed system

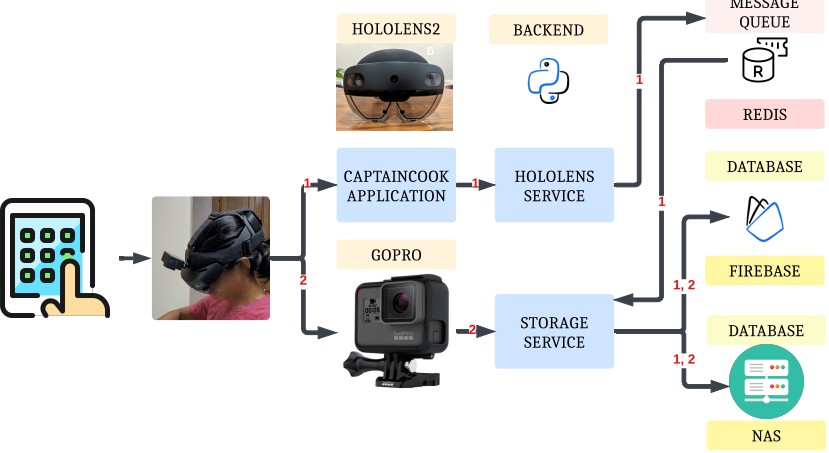

Figure 21: **Architecture** for data capturing system

In developing our data-capturing application, we have utilized data streams from various devices, specifically Hololens2 and a GoPro. The Hololens2 is particularly suited for our needs when set in research mode. It offers a wealth of data from an array of sensors, including a depth sensor, three Inertial Measurement Unit (IMU) sensors - an accelerometer, a gyroscope, and a magnetometer - and spatial information that contains head and hand tracking data. For the Hololens2, we created a custom Unity streamer application, taking inspiration from [14]. This application acts as a server, while

our Python backend application assumes the role of a client. When we initiate a recording session, we establish one TCP socket connection for each sensor to capture data. As the sensor-specific data stream is received, it is immediately pushed onto the sensor-specific Redis message queue [22] Another dedicated Python backend service polls data from these message queues, processes it and subsequently stores it on a locally configured Network Attached Storage (NAS) server. When starting a recording session with GoPro, we utilize the OpenGoPro library to communicate and capture data at the established 4K resolution and 30 FPS. The recorded video is then downloaded from the GoPro through WiFi and saved onto the local NAS server. This architecture, as illustrated in Figure 21, enables us to capture, process, and securely store vast amounts of data in real-time.

### C.1.3 Whom to record

**Participant Statistics.** The statistics concerning the participants who engaged in cooking activities are presented in Figure 22. It is important to highlight that participation in the recording process was entirely voluntary, and participants received no compensation.

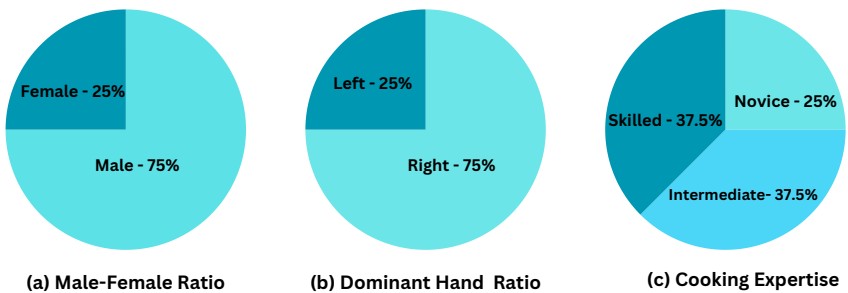

(a) Male-Female Ratio    (b) Dominant Hand Ratio    (c) Cooking Expertise

Figure 22: **Participant Statistics.** Displays information about the participants.

**Participant Training.** To guarantee precise data collection on cooking errors, it is essential that participants have fundamental culinary skills and thorough knowledge of the recipes they will be preparing. To assist participants, we provided them with a comprehensive list of instructional videos on basic culinary skills and techniques specific to different recipes.

**Sources of bias.** We recognize the inherent biases of this dataset, notably the smaller number of participants compared to traditional, large-scale datasets used for action or activity understanding. It is important to note that each participant was required to perform and record the same recipe four times. With each iteration, the recording script was altered, ensuring that each recording remained distinct. Additionally, while many errors were intentionally induced by following a script, participants also made numerous unintentional errors, which they later annotated.

---

[22]https://redis.com/

## C.2 Data Collection

After determining what, where, and whom to record, we began collecting data from participants engaged in cooking activities. Over a period of 32 days, we conducted recordings in 10 different kitchen settings across the United States. Participants were able to schedule their availability for these activities in various kitchen environments. We provide statistics for each selected recipe, detailing the number of normal and error recordings and their respective durations in Table 17 and Figure 23.

Table 17: **Statistics.** $\mathcal{N}_n$–Count of normal recordings taken for the recipe, $\mathcal{D}_n$–Total duration of these normal recordings, $\mathcal{N}_e$–Count of error recordings taken for the recipe, $\mathcal{D}_e$–Total duration of these error recordings.

| Recipe | Steps | $\mathcal{N}_n$ | $\mathcal{D}_n$ (hrs) | $\mathcal{N}_e$ | $\mathcal{D}_e$ (hrs) |
|---|---|---|---|---|---|
| Pinwheels | 19 | 4 | 0.72 | 8 | 1.2 |
| Tomato Mozzarella Salad | 9 | 11 | 1.31 | 7 | 0.64 |
| Butter Corn Cup | 12 | 6 | 1.62 | 8 | 1.49 |
| Tomato Chutney | 19 | 7 | 3.34 | 8 | 2.01 |
| Scrambled Eggs | 23 | 6 | 2.69 | 10 | 3.13 |
| Cucumber Raita | 11 | 12 | 2.9 | 8 | 1.36 |
| Zoodles | 13 | 5 | 1.35 | 10 | 2.19 |
| Microwave Egg Sandwich | 12 | 6 | 1.05 | 12 | 1.67 |
| Sauted Mushrooms | 18 | 6 | 2.73 | 8 | 2.21 |
| Blender Banana Pancakes | 14 | 7 | 1.78 | 12 | 2.57 |
| Herb Omelet with Fried Tomatoes | 15 | 6 | 1.73 | 11 | 2.14 |
| Broccoli Stir Fry | 25 | 11 | 5.74 | 5 | 1.68 |
| Pan Fried Tofu | 19 | 8 | 3.38 | 7 | 2.31 |
| Mug Cake | 20 | 7 | 2.44 | 10 | 2.32 |
| Cheese Pimiento | 11 | 6 | 1.47 | 9 | 1.72 |
| Spicy Tuna Avocado Wraps | 17 | 7 | 2.0 | 11 | 2.66 |
| Caprese Bruschetta | 11 | 6 | 1.92 | 12 | 2.73 |
| Dressed Up Meatballs | 16 | 6 | 2.0 | 10 | 3.09 |
| Microwave Mug Pizza | 14 | 7 | 1.47 | 6 | 1.14 |
| Ramen | 15 | 10 | 2.40 | 7 | 1.45 |
| Coffee | 16 | 8 | 1.97 | 7 | 1.58 |
| Breakfast Burritos | 11 | 6 | 1.22 | 10 | 1.52 |
| Spiced Hot Chocolate | 7 | 6 | 0.82 | 10 | 1.01 |
| Microwave French Toast | 11 | 9 | 1.94 | 5 | 0.66 |
| **Total** | 384 | 173 | 50.05 | 211 | 44.41 |

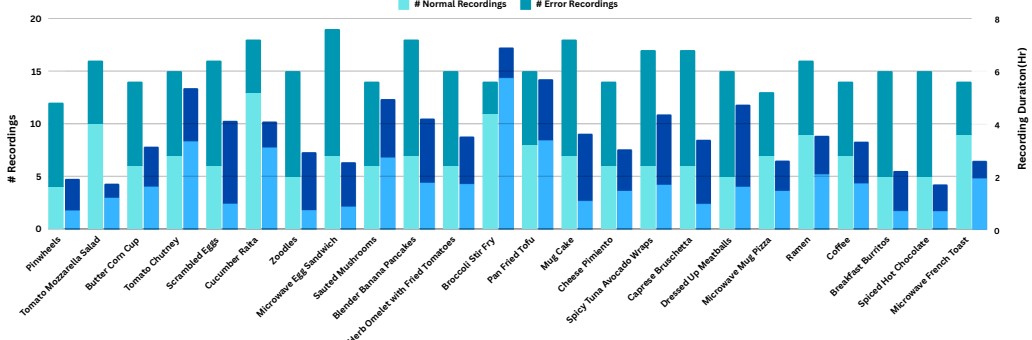

Figure 23: **Statistics.** Count and duration (in hours) statistics of normal and error recordings.

**Inspection & Acclimatisation.** Before initiating the recording process in each environment, participants followed a series of preparatory steps. Initially, they were instructed to remove any identifiable information from body parts visible during the recording. Additionally, they were checked to ensure they were not carrying personal identification devices, such as smartwatches containing personal data. As participants were operating in unfamiliar kitchen settings, they received a comprehensive orientation on the locations of all essential ingredients needed to complete the recipe.

**Normal Recordings.** Participants were provided with a tablet to access the user application described earlier. Initially, they were instructed to perform normal activities. Upon choosing a normal activity, each participant was presented with a sequence of steps that followed the topological order of the action-centric task graph constructed for that activity. Participants were expected to adhere strictly to the sequence displayed on the tablet and avoid any deviations that could lead to errors. However, participants committed numerous unintentional errors during their first execution of any given recipe and later annotated the errors induced accordingly.

**Error Recordings.** We developed three strategies[23] for participants to choose from, each tailored to perform the recipe in a specific environment. After choosing the strategy, participants were given detailed instructions on how to perform the recipes. We list the strategies presented to the participants (1) **Impromptu**: Participants were asked to induce errors while performing the recipe. Following the completion of each recording, participants used a web-based interface to update the errors they performed during each step. Due to the complex nature of cooking activities and the lack of experience of the participants in cooking, many errors induced in this strategy were **unintentional**. (2) **Disordered Steps**: Participants were given pre-prepared error scripts with missing steps and ordering errors. (3) **Induct Error**: Participants used a web-based interface to create an error script for each selected recipe recording. The modified recipe steps were displayed on a tablet, enabling participants to perform according to their scripted errors.

**Caveats.** We rely on a tablet-based interface to display the sequence of steps and also capture recordings in 4K resolution. Thus, we are aware that an OCR-based system can recognize the active step information in the tablet. To address this, we made sure that the test set included videos in which participants viewed the entire recipe instruction text as a paragraph instead of a sequence of steps.

---

[23]The practice of using scripted videos for activity understanding [82] has inspired us to develop the strategies.

## C.3 Data Processing

### C.3.1 Synchronization

After recording sessions in a kitchen environment, the data is transferred to a local NAS system and a synchronization service is run to align the raw data streams captured by the Hololens2. This includes synchronizing data from multiple streams—RGB, depth, spatial, and three Inertial Measurement Unit (IMU) sensors—using timestamps provided by the Hololens2. Post synchronization, both the raw and synchronized data are uploaded to cloud storage, and the links are made public.

### C.3.2 Annotation

**Coarse-Grained Action/Step Annotations.** We developed an interface for performing step annotations in Label Studio[24]. This interface is used by each annotator to mark the start and end times of each step. Our steps are significantly longer than individual fine-grained actions and encompass multiple fine-grained actions required to perform the described step. Table 18 presents a summary and comparison of coarse-grained action/step annotations for our dataset alongside other popular datasets. To facilitate these annotations, we used both our user application and Label Studio. We integrated our application with Label Studio through its provided APIs, enabling the seamless creation of a labelling environment for each recording and ensuring that annotations are reliably stored.

Table 18: Comparison of coarse-grained action or step annotations across related datasets. Here, $\mathcal{T}_{avg}$ represents the avg. duration for each video, $\mathcal{N}^{seg}$ shows the total number of segments, $\mathcal{N}^{seg}_{avg}$ reveals the avg. number of segments per video, and $\mathcal{T}^{seg}_{avg}$ shows the avg. duration for all segments.

| Dataset | $\mathcal{T}_{avg}$ (min) | $\mathcal{N}^{seg}$ | $\mathcal{N}^{seg}_{avg}$ | $\mathcal{T}^{seg}_{avg}$ (sec) |
|---|---|---|---|---|
| 50Salads | 6.4 | 899 | 18 | 36.8 |
| Breakfast | 2.3 | 11,300 | 6.6 | 15.1 |
| Assembly 101 | 7.1 | 9523 | 24 | 16.5 |
| CSV | 0.2 | 18488 | 9.53 | 2.1 |
| HoloAssist | 4.48 | 15927 | 7.17 | 39.3 |
| **Ours (Total)** | **14.8** | 5300 | 13.8 | **52.78** |

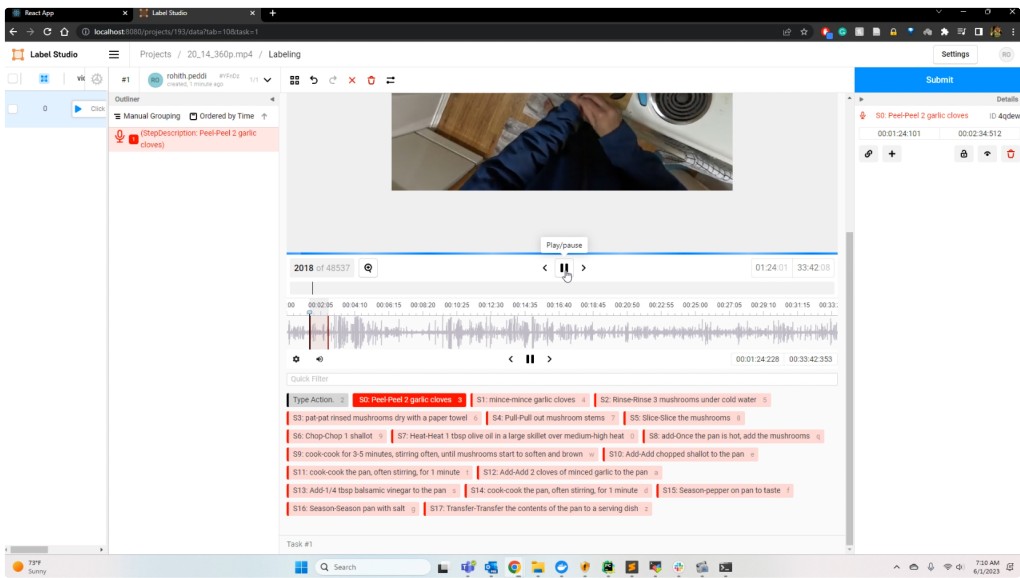

Figure 24: **Annotation Interface** developed to generate step annotations for a recording

**Annotation Interface.** We will briefly explain the rationale behind the design choices for our annotation interface. First, in step annotations, our goal is to define the temporal boundaries for each step of the recording. Consequently, we have positioned a complete list of all the steps associated with

---

[24]https://labelstud.io/

the activity beneath the video. When a time period is identified as the boundary for a specific activity step, it appears on the left-hand side of the screen. Simultaneously, the start and end times of the step are displayed on the right side in the corresponding time slots, allowing for minor adjustments.

**Fine-Grained Action Annotations.** Drawing inspiration from the pause-and-talk narrator [11], we have developed a web-based tool for fine-grained action annotation that leverages OpenAI's Whisper APIs for speech-to-text translation. While this system is built around the Whisper API, it is designed to be flexible enough to integrate any automatic speech recognition (ASR) system capable of handling transcription requests. Upon its acceptance, we will release this web-based annotation tool. Figure 25 illustrates the key steps for a recording and their corresponding step and action annotations.

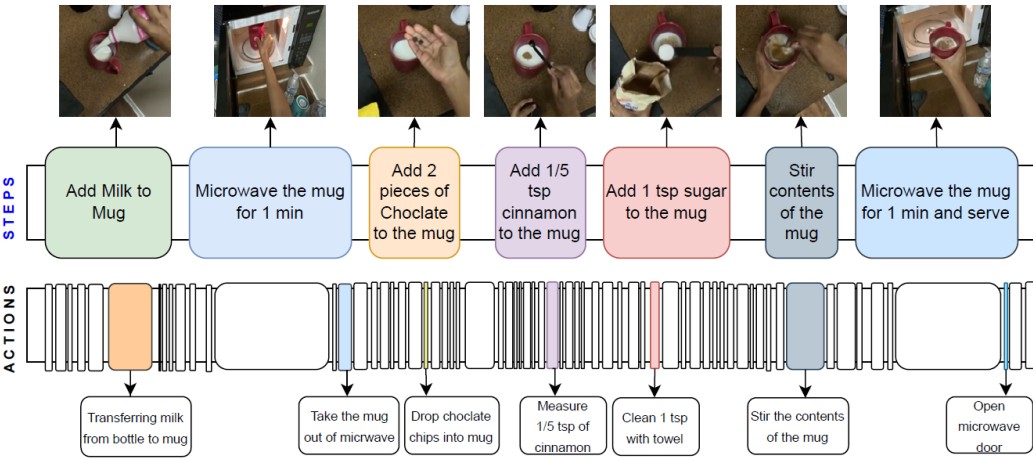

Figure 25: **Step and action annotations.** for the recipe *Spiced Hot Chocolate*.

**Error annotations.** Participants are required to document the errors (Appendix C.2) made during each recording. We compile the error descriptions and categorizations provided by the participants and succinctly display them, as shown in Figure 3 (see Error Categories) in the main paper. In Figure 26, we present the frequency of each error category type induced during execution.

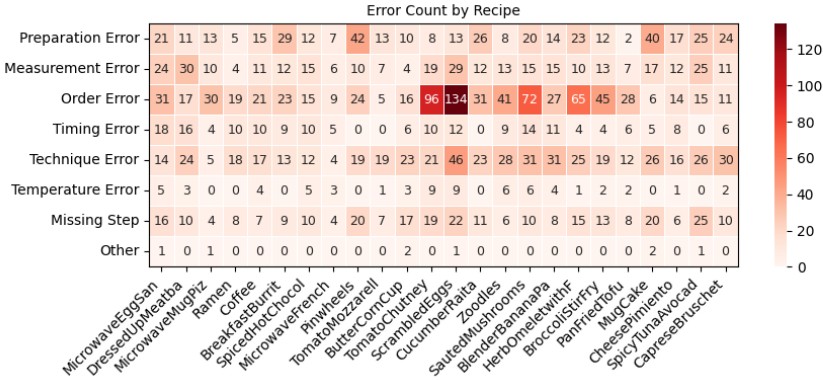

Figure 26: **Frequency of errors.** induced in the recordings for each recipe type.

### C.3.3 Data Composition

In this section, we list down all the components provided as part of our data. **Raw and synchronized multi-modal data from Hololens2:** The dataset includes raw data captured using the Hololens2 device. This data is multi-modal, which means it contains information in several different forms,

including visual (e.g., images or videos), auditory (e.g., sounds or speech), and others (like depth information, accelerometer readings, etc.). **4K videos from GoPro:** Includes high-resolution 4K videos recorded using a GoPro camera. Such high-resolution video can provide much detail, particularly useful for tasks like object recognition. **Step annotations** for all the data. **Fine-grained actions for 20% of the data:** Fine-grained actions might include specifics about what objects are being manipulated, exactly what movements are being made, and so on. This data could be helpful for tasks that involve understanding or predicting specific types of actions. **Extracted features using multiple backbones for different tasks:**. We provide a comprehensive overview of the components we release with the dataset in table 19

Table 19: This table presents an overview of the components we release as part of the dataset.

| | | | | |
|---|---|---|---|---|
| Hololens2 | Raw | RGB
RGB pose
Depth
Depth pose
Audio
Head pose
Left wrist pose
Right wrist pose
IMU Accelerometer
IMU Gyroscope
IMU Magnetometer | Synchronized | RGB
RGB pose
Depth
Depth pose
Audio
Head pose
Left wrist pose
Right wrist pose
IMU Accelerometer
IMU Gyroscope
IMU Magnetometer |
| Gopro | Raw | RGB
Audio | | |
| Annotations | Step
Fine-grained Action | | | |

### C.3.4 Maintenance

The dataset will be hosted on Box data storage drives and accessible via a publicly available link. The associated website will provide information about the code, dataset, and other details.

### C.3.5 License

Copyright [2023] [The University of Texas at Dallas]

Licensed under the Apache License, Version 2.0 (the "License"); you may not use this file except in compliance with the License. You may obtain a copy of the License at

http://www.apache.org/licenses/LICENSE-2.0

Unless required by applicable law or agreed to in writing, software distributed under the License is distributed on an "AS IS" BASIS, WITHOUT WARRANTIES OR CONDITIONS OF ANY KIND, either express or implied. See the License for the specific language governing permissions and limitations under the License.

