# OpenReview forum: "CaptainCook4D: A Dataset for Understanding Errors in Procedural Activities"
_NeurIPS.cc/2024/Datasets_and_Benchmarks_Track — NeurIPS 2024 Track Datasets and Benchmarks Poster_

### Official Review · Reviewer_4sjU · 2024-07-22
**A good dataset for understanding errors in procedural activities.**

**Rating:** 8
**Confidence:** 4
**Correctness:** Yes.
**Clarity:** Yes.

**Review:**

The quality, clarity, originality and significance of this work are good.
Pros:
1. The article is a well-written.
2. The dataset is well-designed.
3. The experiments are well-designed.

Cons:
1. In related work, the scientific progress in the research domains (error recognition, temporal action localization, procedure learning) has not been discussed.
2. Without clear fine-grained and coarse-grained action definitions, it is difficult for your action annotations to have consistency.
3. The experiment of procedure learning seems too simple to answer the question Q4 sufficiently.

**Strengths:**

1. Error Inclusion: CaptainCook4D incorporates both normal and error scenarios, enabling the development of AI systems capable of recognizing and anticipating errors.
2. Comprehensive Annotations: The dataset provides detailed annotations.
3. Benchmarking: The paper benchmarks the dataset for tasks such as error recognition, multi-step localization, and procedure learning, providing baseline models and evaluation metrics.

**Additional Feedback:**

There is no further feedback.

**Documentation:**

Yes.

**Ethics:**

No.

**Limitations:**

Yes.

**Opportunities For Improvement:**

1. In related work, provide a brief literature review on error recognition, temporal action localization, procedure learning.
2. Define your fine-grained and coarse-grained actions, and make sure the consistency of your annotations.
3. Benchmark more procedure learning methods.
4. For each benchmarked task, a brief discussion on how it can be used in real-world applications is suggested.

**Relation To Prior Work:**

Yes.

**Summary And Contributions:**

This paper introduces CaptainCook4D, a new dataset designed to study errors in procedural activities, particularly in cooking. The dataset consists of 384 recordings (94.5 hours) of individuals executing recipes in real kitchen settings, capturing both intentional and unintentional deviations from instructions. The authors benchmarked the dataset for three tasks: : error recognition, multi-step localization and procedure learning.

---

> ### Author Rebuttal · Authors · 2024-08-15
>
> > The quality, clarity, originality and significance of this work are good
>
> Thanks a ton for your encouraging comments! Your discussion indeed offers valuable ideas for enhancing the paper's content.
>
> ----
> > The scientific progress in the research domains (error recognition, temporal action localization, procedure learning) has not been discussed.
>
> In the current version of the paper that is under review, this section was placed in the supplementary material due to space constraints in the main paper. For the camera-ready version, we shall relocate it to the main paper.
>
> ----
> > Define your fine-grained and coarse-grained actions and make sure of the consistency of your annotations.
>
> Following Assembly101 and EPIC-Kitchens, we have classified actions into two categories: fine-grained and coarse-grained actions, which are illustrated in Figure 18 in Appendix C. We define the categories as follows:
> - **Fine-grained actions.** consists of atomic operations like cutting, chopping, opening, and closing. We provide annotations marking the start and end times for each of these actions.
> - **Coarse-grained actions.** describe individual steps within a recipe, encompassing a series of basic operations that collectively achieve a specific objective. Each of these actions is also annotated with precise start and end times.
>
> ----
> > The experiment of procedure learning seems too simple to answer question Q4 sufficiently.
>
> Leveraging our annotations (which include task graphs of recipes and coarse-grained actions for recordings), we focused on addressing the question (Q4): *Can we infer the underlying procedure (recipe text) from the videos of a particular recipe under different levels of supervision (i,e supervised, weakly-supervised and self-supervised procedure learning variants)?*.
>
> In the version of the paper under review, we opted to explore the more challenging variant of procedure learning \textemdash self-supervised procedure learning. This approach aims to deduce the procedure directly from a given set of videos corresponding to a procedural activity. To accomplish this, we employed two state-of-the-art (at the time of writing) self-supervised procedure learning methods and presented results in Table 6 and Appendix B.6.
>
> Additionally, we wish to highlight the recent contribution by [1], in which the authors proposed a novel method for supervised procedure learning. We are also exploring Neuro-Symbolic techniques for weakly supervised procedure learning.
>
> ----
> > Benchmark more procedure learning methods.
>
> In future versions of the project, we aim to expand our baselines and plan to include baselines for (a) weakly supervised procedure learning and (b) out-of-distribution detection in procedural activity videos (which can be viewed as error recognition from a distributional lens).
>
>
> ----
> > For each benchmarked task, a brief discussion on how it can be used in real-world applications is suggested.
>
> While not exhaustive, we have identified and listed several real-world applications related to our benchmark tasks.
>
> - **Surveillance Systems:** By integrating benchmark tasks of multi-step localization and error recognition, we can establish a basic surveillance system for monitoring procedural activities.
> - **Guidance Systems:** Our zero-shot error recognition can be used to identify procedural errors in real-time, providing immediate feedback during activities.
> - **Skill Assessment Systems:** Our tasks focused on supervised error recognition can leverage multimodal data to evaluate the skill levels of individuals engaged in procedural activities.
> - **Quality Control Systems:** Our benchmark tasks, particularly error recognition, can be integrated into quality control systems within manufacturing or culinary industries to automatically detect deviations from standard procedures, ensuring consistent product quality.
> - **Multi-modal Auditing Systems:** The procedural data can be used to automatically generate documentation of the steps involved in a task. Our benchmark tasks on procedure learning can be useful for building such systems.
>
> -----
> [1] Differentiable Task Graph Learning: Procedural Activity Representation and Online Mistake Detection from Egocentric Videos

---

> > ### Comment · Reviewer_4sjU · 2024-08-23
> >
> > The authors have cleared my doubts. I will keep my positive score.

---

### Official Review · Reviewer_jiMN · 2024-07-24
**CaptainCook Review**

**Rating:** 7
**Confidence:** 3
**Clarity:** Yes!

**Review:**

This dataset and benchmark of errors in cooking videos is a high-quality and clear contribution.

It lacks originality, as it follows in the footsteps of prior work on error quantification in recipes, but it is significant in that it expands the data and taxonomy available to methods that aim to classify errors. It is also significant in the usefulness to real-world agents of understanding error failure modes.

pros:
- 100 hours of data
- nice taxonomy of various different types of error
- comprehensive baselines and evaluations for the usefulness of the dataset at teaching about errors

cons
- only 8 participants
- only 10 kitchens
- zero shot methods and baselines only achieve between 40-70% accuracy on error detection --> also a pro as more work to be done
- multi-modality won't exist for all data, however it turns out this barely helps vs. just better models that are trained with visual data
- not enough analysis of how modalities help/have no effect for understanding errors
- lack of discussion of anomaly detection and comparison to those methods as a basis for identifying errors
- lack of comparison to models trained on prior work as a way of understanding the utility of the new dataset

**Strengths:**

Understanding failure modes is the primary blocker for deploying autonomous systems into in-the-wild scenarios such as real-world kitchens and/or factories. This paper does an excellent job at presenting a number of promising error classifications and examples of such errors and providing baselines for recognizing these errors.

The paper does a tremendous job of evaluating a series of baselines on their error dataset. They include 3 (4) baselines across a variety of modalities such as features / video / audio / depth / text.

**Additional Feedback:**

N/A

**Correctness:**

The dataset seems sound and well put together, despite the inherent limitations in quantifying the expansive set of possible negative errors that can occur. The experimental design is comprehensive and tests a number of interesting features with regard to performance on error detection.

**Documentation:**

Yes!

**Ethics:**

The only concern I'm flagging is that this work already appeared in a prior workshop at ICML 2023 and I'm not sure if that precludes it from participating in the Neurips D&B track.

Other concerns are about privacy of individuals in their kitchen, but an IRB was consulted and users were told to remove identifying information from their kitchens.

**Limitations:**

The authors are releasing a dataset with only ~100 hours of video, given that the scale of data that models train with in 2024 is quite large, it might be insufficient to learn a more generalizable/emergent understanding of errors.

There are a number of pre-existing datasets that include errors and analyze their types. The related work section here considers those adequately, but could do better in comparing to them.

**Opportunities For Improvement:**

The dataset covers 24 recipes. It also includes 8 participants in 10 kitchens. It argues that it is a real-life setting and not a lab setting, but with only 10 kitchens and 8 participants, it probably shares a lot of similarities to a limited lab environment.

The authors could include a broader diversity of participants and kitchens to approach a more realistic, less curated environment.

**Relation To Prior Work:**

Yes, there is a useful table comparing against prior error datasets (and other relevant datasets). However, it would have been helpful if the paper included a comparison of how backbones trained on prior datasets compared to backbones trained on their dataset.

**Summary And Contributions:**

This paper contributes almost 100 hours across 384 recordings of kitchen activities. These kitchen activities follow a procedure or recipe. Some of them are recorded under the instruction of making a mistake. These mistakes are annotated and analyzed, and the paper explores this negative space of failed steps, unlike most prior work.

The dataset covers 24 recipes. It also includes 8 participants in 10 kitchens

The dataset includes errors that occurred in 3 different ways. These are:

1) Impromptu errors
2) Disordered steps
3) Induct errors

Each of these groups of errors can have different manifestations, and lead to a broader set of possible errors.

The larger taxonomy of errors is:

1) Preparation errors
2) Technique errors
3) Measurement errors
4) Temperature errors
5) Order errors
6) Timing errors
7) Missing steps

Each of these 7 errors occurs up to ~300 times on average in the dataset, with temperature errors occuring only 66 times and order errors occuring 795 times (from the earlier step 2).

The paper also includes a set of baselines designed to recognize and/or classify these errors in videos as/when they occur. The baselines are very up-to-date including pre-trained VLMs and comparing with a number of backbones.

The authors test these backbones on two splits, one that splits up recordings into steps and the other that splits up recordings into train/val/test based on which video is being used.

The paper also contributes a zero shot baseline that uses knowledge about the recipe graph as a way of generating prompts to use with a VLM.

---

> ### Author Rebuttal · Authors · 2024-08-15
>
> > This paper does an excellent job at presenting a number of promising error classifications ..... The paper does a tremendous job of evaluating a series of baselines
>
> Thank you for your encouraging comments and an excellent summary of our work. Your discussion has helped improve the presentation of the proposed dataset and its associated challenges. Thanks a ton!
>
> ----
> > The only concern I'm flagging is that this work already appeared in a prior workshop at ICML 2023, and I'm not sure if that precludes it from participating in the Neurips D\&B track.
>
> Our submission to a **non-proceedings track** DMLR workshop at ICML 2023 does not preclude it from participating in the NeurIPS D&B track in 2024 for the following reasons:
> - **Rules.** According to the guidelines on the NeurIPS website:
>   - **Dual Submission Policy.**  Submissions that are substantially similar to papers that the authors have previously published or submitted in parallel to other peer-reviewed venues with proceedings or journals may not be submitted to NeurIPS. **Papers previously presented at workshops are permitted, so long as they did not appear in conference proceedings (e.g., CVPRW proceedings), a journal or a book.**  NeurIPS coordinates with other conferences to identify dual submissions.  The NeurIPS policy on dual submissions applies for the entire duration of the reviewing process.  Slicing contributions too thinly is discouraged.  The reviewing process will treat any other submission by an overlapping set of authors as prior work. If publishing one would render the other too incremental, both may be rejected.
>   - **Non-Proceedings Track Submission.** Our earlier submission of the preliminary version of the dataset to the DMLR workshop at ICML 2023 was under a non-proceedings track, meaning it did not appear in any form of proceedings (conference/journal/book).  Therefore, we want to clarify that, based on the Dual Submission Policy mentioned above, our previous workshop submission does not in any way preclude our work from participating in the NeurIPS D&B track 2024.
> - **Submission Content.** It is crucial to emphasize that the version currently under review for the NeurIPS D\&B track is significantly different from our previous workshop submission.
>
> ----
> > zero shot methods and baselines only achieve between 40-70\% accuracy on error detection --> also a pro as more work to be done
>
> - **Observations.** Our experiments revealed that both supervised and zero-shot error recognition methods struggle to **accurately interpret observed events**.
> -  **Way Forward.**
>    - Error Recognition is a challenging task, and we conjecture that it requires the development of techniques that understand the context, meaning, and cause of various errors.
>    - We note that integrating more effective perception modules can achieve initial improvements in both tasks (zero-shot and supervised error recognition). However, to further advance the field of error understanding, we require a sophisticated reasoning system that can discern between what is observed and what is expected.
>    - To this end, we are building an enhanced reasoning system equipped with common-sense knowledge from Large Language Models (LLMs) to effectively differentiate observed from expected outcomes in procedural activities.
>
> ----
> > multi-modality barely helps vs. just better models that are trained with visual data ....  analysis of how modalities help/have no effect on understanding errors
>
> Good point! We summarize our findings and recommendations below and shall add them to the camera-ready version.
>
> - **Observations.**
>   - By utilizing multimodal features extracted using Imagebind to train our models, we observed a significant increase in predictive performance compared to models trained exclusively using features extracted using RGB video data (Table 2).
>   - When we compared models trained with multimodal features from Imagebind against those using video features from various pre-trained video recognition models, the performance varied significantly depending on the chosen pre-trained model. In some instances, models trained using only video features outperformed those trained using multimodal features.
>   - We also observed that the model trained using video features extracted with ImageBind performed worse than those trained using video features from other pre-trained models.
> - **Strategies for Improvement.**
>   - Based on the observations above, we can conclude that multimodal features significantly enhance the predictive performance of trained models. However, the extent of improvement relies heavily on the specific choice of pre-trained models and engineering components employed to extract multimodal features.
>   - We would like to emphasize that our approach to using multimodal data sets a foundational benchmark. We encourage the research community to explore more sophisticated architectures for processing multimodal data.
>
> ----
> > lack of discussion of anomaly detection and comparison to those methods as a basis for identifying errors
>
> We summarize our findings from Appendix B and present further recommendations below, which will be included in the camera-ready version.
>
> - **Observations.**
>   - We experimented with two state-of-the-art (at the time of writing) self-supervised anomaly detection methods and presented our analysis in Appendix B.
>   - We noticed that these methods performed close to random, and we attribute their extremely low performance to the traditional anomaly detection setups designed primarily for exocentric videos with close to minimal camera movement.
> - **Strategies for Improvement.** Based on the above observations, we conclude that self-supervised anomaly detection in egocentric videos, particularly through one-class classification, is extremely challenging due to the lack of static cameras. Thus alluding to the need for further exploration using more sophisticated anomaly detection methods, beginning with supervised approaches.

---

> > ### Author Rebuttal · Authors · 2024-08-15
> >
> > > lack of comparison to models trained on prior work as a way of understanding the utility of the new dataset
> >
> > If we've understood your concern correctly, you're referring to the absence of a specific type of comparison in which, for a given task, the performance of a specific method on our dataset is compared to the method's performance on similar new datasets that include errors.
> >
> > At the time of writing, such a comparison wasn't feasible due to the choice of benchmark tasks. Error Recognition as a task is novel, and most prior works that developed datasets with errors have established their benchmarks by treating the problem as a supervised classification task (specifically training an MLP using features extracted from different pre-trained backbones). Similarly, our work adopts this approach and presents results on supervised binary classification tasks. Additionally, we also introduce a novel zero-shot error recognition baseline variant using a prompt-and-predict strategy. Finally, tasks such as multi-step localization and self-supervised procedure learning have not been used as benchmarks in other datasets.
> >
> > We also want to highlight a recent contribution by [1], which introduced a novel supervised procedure learning baseline on our dataset. We hope this addresses your concern; if you still need to, please provide further details about your query.
> >
> > ----
> > > The authors could include a broader diversity of participants and kitchens to approach a more realistic, less curated environment.
> >
> > Excellent point! We intend to extend our dataset as an egocentric analogue to the CrossTask dataset. We aim for it to serve as a platform for testing novel algorithms designed for various procedural learning tasks as well as assess the robustness of these algorithms to distribution shifts induced by different categories of errors.
> >
> > To this end, considering the feedback from reviewers, we are investigating additional relevant datasets for extension. We aim to enhance our dataset with data from several other interesting domains and benchmark tasks that focus on transfer learning across these domains.
> >
> > ----
> > > The authors are releasing a dataset with only ~100 hours of video; given that the scale of data that models train within 2024 is quite large, it might be insufficient to learn a more generalizable/emergent understanding of errors.
> >
> > Error Understanding in procedural activities is an extremely challenging task and demands developing a system that equips an efficient perception system and a robust reasoning system. Below, we list the challenges associated with constructing such a system:
> >
> > - **Data.**
> >   - As highlighted by the reviewer, more data can enhance our understanding of errors. However, unlike action recognition or object detection, acquiring non-redundant data corresponding to errors (that occur while performing procedural activities) is an extremely costly process.
> >   - Capturing large-scale procedural activity data with errors also involves significant logistical challenges, demanding the collaboration of various teams from different regions (similar to the Ego-Exo4D project). Despite its relatively small size, our dataset is unique in that it encompasses both intentional and unintentional errors, offering a wider variety of errors than typically seen in other datasets.
> >   - To this end, following the suggestions from reviewers, we are exploring building the next version of our dataset by extending it to include videos from several other interesting domains.
> > - **Algorithms.**  Understanding errors in procedural activities necessitates integrating an efficient perception system capable of accurately interpreting observed events with a reasoning system. This reasoning system must possess essential common-sense knowledge about the activity to distinguish between what is observed and what should occur.
> >
> > ----
> > > The related work section here considers those adequately but could do better in comparing them.
> >
> > We shall incorporate this change in the camera-ready version of the paper.
> >
> > -----
> >
> > [1] Differentiable Task Graph Learning: Procedural Activity Representation and Online Mistake Detection from Egocentric Videos

---

> > ### Comment · Area_Chair_UdvX · 2024-08-16
> >
> > Dear Authors and Reviewers,
> >
> > I confirm that the paper does not qualify as a dual submission as the mentioned workshop did not publish any proceedings.
> >
> > Reviewers and Authors please continue the discussion on the paper's merits.

---

### Official Review · Reviewer_wKvX · 2024-07-25
**A new dataset for error understanding**

**Rating:** 8
**Confidence:** 4
**Correctness:** Yes
**Clarity:** Yes

**Review:**

The paper introduces a novel  dataset aiming at understanding errors in procedural activities, particularly focusing on cooking tasks. The dataset contains 384 recordings totaling 94.5 hours, with participants either following recipe instructions accurately or intentionally and unintentionally deviating from them to induce errors. The authors provide detailed annotations for steps and actions, as well as descriptions of errors, facilitating research on error recognition, multi-step localization, and procedure learning.

Pros:
1. The creation of this dataset with a focus on errors in cooking activities is a valuable contribution to the field.
2. Providing step and action annotations along with error descriptions is unique and can help the research in various tasks.

Cons:
1. The dataset is focused on cooking activities, which may limit its applicability to other domains.
2. Data collection in a specific geographic area and with a limited number of participants could introduce regional or demographic biases.
3. The errors captured represent a subset of all possible errors, which may not fully capture the complexity of real-world mistakes.

**Strengths:**

1. The creation of this dataset with a focus on errors in cooking activities is a valuable contribution to the field.
2. Providing step and action annotations along with error descriptions is unique and can help the research in various tasks.

**Additional Feedback:**

N/A

**Documentation:**

Yes

**Limitations:**

Yes

**Opportunities For Improvement:**

The authors may consider increasing the dataset coverage, by annotating existing goal-oriented egocentric video data from datasets like Ego-Exo4D and EgoExoLearn. This can help increase the demographic diversity of this work.

**Relation To Prior Work:**

Yes. Several very recent works can be also discussed: Ego-Exo4D, EgoExoLearn, EgoTaskQA. All of these also contain many cooking scenarios.

**Summary And Contributions:**

This dataset focuses on understanding the procedural activities in videos. A total of 94.5 hours of videos are collected, containing people cooking in the kitchens. This dataset consists of two types of activities: one in which participants adhere to the provided recipe instructions and another where they deviate and induce errors.  5.3K step annotations and 10K action annotations are provided.  Error recognition, multi-step localization, and procedure learning are used for benchmarking.

The major contribution is that, this dataset contains error performs and their annotation, which I believe is unique to the field.

---

> ### Author Rebuttal · Authors · 2024-08-15
>
> > (a) The creation of this dataset with a focus on errors in cooking activities is a valuable contribution to the field \ldots (b) Providing step and action annotations along with error descriptions is unique and can help the research in various tasks.
>
> Thank you very much for your valuable suggestions! Your comments indeed offer several strong directions to enhance the proposed dataset.
>
> ----
>
> > Data collection in a specific geographic area and with a limited number of participants could introduce regional or demographic biases.
>
> - **Demographic Bias.** Capturing 4D data on errors during procedural activities in real-world kitchens posed significant logistical issues. Consequently, we restricted data collection to a specific geographical area while ensuring significant diversity in the recording environments. We will include these details along with the additional information requested by the Ethics Reviewer in the supplementary section of the camera-ready version of the paper.
> - **Participant Bias.** Our dataset presents the first attempt at building a comprehensive 4D dataset to study errors in procedural tasks. We acknowledge that the dataset exhibits inherent biases due to the relatively small number of participants compared to conventional, large-scale datasets for action or activity understanding. However, it's important to note that while each participant performs and records the same recipe four times, each recording is unique because the script changes (occasionally incorporating errors) every time. This approach ensures variability within the data collected.
>
> **Dataset Extension.** Following recommendations from reviewers, we are investigating the possibility of including carefully sampled data from (a) Other cooking datasets and (b) Other datasets developed over different domains.
>
> ----
> > The errors captured represent a subset of all possible errors, which may not fully capture the complexity of real-world mistakes.
>
> - **Combinatorial Nature.** Correct! Our dataset captures only a subset of all potential errors. Nevertheless, it's important to emphasize that the range of errors possible during procedural activities is inherently combinatorial. Consider the seemingly simple act of opening a container. While the basic actions (pick up, twist lid, place down) remain consistent, the potential errors are numerous and diverse. These may include applying excessive force, using incorrect tools, mishandling fragile containers, contamination, inadequate grip, incorrect opening of containers leading to spills, and many more.
> - **Way Forward.**
>   - **Error Understanding.** (owing to the combinatorial nature of errors) demands a more nuanced analysis where the developed AI systems should focus on efficiently differentiating observed events from expected outcomes. Thus, our dataset, with its inclusion of a wide spectrum of potential errors from different categories, offers an excellent test bed and a comprehensive starting point to validate the efficacy of AI systems in recognizing errors.
>   - **Procedural Tasks.** Traditionally, methods developed for procedural tasks primarily focused on predictive performance over datasets composed solely of in-distribution samples (e.g., normal recordings without errors in our dataset). Our dataset carefully includes a sufficient number of recordings for each step that do not possess any errors (in-distribution samples; normal recordings in our dataset) alongside recordings that captured errors (out-of-distribution samples; error recordings in our dataset). Therefore, our dataset is also well-suited as a test bed for assessing the robustness of methods developed for several procedural tasks.
> - Our benchmark tasks, specifically **Zero-Shot Error Recognition** and **Robust Multi-Step Localization**, exemplify the types of tasks we are focusing on as mentioned above.
>
> ----
> > The authors may consider increasing the dataset coverage by annotating existing goal-oriented egocentric video data from datasets like Ego-Exo4D and EgoExoLearn. This can help increase the demographic diversity of this work.
>
> Thank you for the excellent suggestion! Following the reviewer's recommendation, we examined videos from Ego-Exo4D but discovered only a few that were suitable. Luckily, we found several interesting and useful videos from EgoExoLearn to incorporate into our dataset. Going forward, we will diligently review available datasets, sample additional videos, annotate them and integrate them into our dataset to increase its diversity.
>
> ----
> > The dataset is focused on cooking activities, which may limit its applicability to other domains.
>
> We agree that with a specific focus on cooking activities, our dataset cannot directly be used for pre-training models and using them in a zero-shot way for other domains. However, cooking activities, with their inherent complexities, share a range of actions with several other domains of interest, such as pour, take, lift, place, twist, etc. Thus, by including long recipes (with more steps in the procedure) from a range of culinary traditions, the research community can potentially sample data corresponding to these actions, include them in their pre-training data, and fine-tune pre-trained models on their respective domains of interest.
>
> **Way Forward.** Following your suggestions, we intend to extend and position this dataset as an ego-centric analogue of CrossTask for procedural understanding tasks in general and error understanding tasks in specific. This will allow future versions of the dataset to be readily applicable for cross-domain tasks.

---

> > ### Comment · Reviewer_wKvX · 2024-08-23
> >
> > Thanks for the rebuttal. As I stated in the initial review, I still believe this is a good paper and should be accepted to this venue.

---

### Decision · Program_Chairs · 2024-09-26

**Decision:**

Accept (Poster)

**Comment:**

The paper received unanimous positive assessments, with the following recommendations: “8: Top 50% of accepted papers, clear accept”, “7: Good paper, accept”, “8: Top 50% of accepted papers, clear accept”.

While reviewers has some concerns prior to the rebuttal, these have been adequately discussed and clarified in the rebuttal, as reviewers confirm.

Overall, it’s clear that the proposed paper has relevance, hence the AC recommends acceptance.